# The relative binding position of Nck and Grb2 adaptors impacts actin-based motility of Vaccinia virus

Angika Basant*, Michael Way*

Cellular Signalling and Cytoskeletal Function Laboratory, The Francis Crick Institute, London, United Kingdom

**Abstract** Phosphotyrosine (pTyr) motifs in unstructured polypeptides orchestrate important cellular processes by engaging SH2-containing adaptors to assemble complex signalling networks. The concept of phase separation has recently changed our appreciation of multivalent networks, however, the role of pTyr motif positioning in their function remains to be explored. We have now investigated this parameter in the operation of the signalling cascade driving actin-based motility and spread of Vaccinia virus. This network involves two pTyr motifs in the viral protein A36 that recruit the adaptors Nck and Grb2 upstream of N-WASP and Arp2/3 complex-mediated actin polymerisation. Manipulating the position of pTyr motifs in A36 and the unrelated p14 from Orthoreovirus, we find that only specific spatial arrangements of Nck and Grb2 binding sites result in robust N-WASP recruitment, Arp2/3 complex driven actin polymerisation and viral spread. This suggests that the relative position of pTyr adaptor binding sites is optimised for signal output. This finding may explain why the relative positions of pTyr motifs are frequently conserved in proteins from widely different species. It also has important implications for regulation of physiological networks, including those undergoing phase transitions.

**\*For correspondence:**
angika.basant@crick.ac.uk (AB);
michael.way@crick.ac.uk (MW)

**Competing interest:** The authors declare that no competing interests exist.

## Editor's evaluation

The authors have previously established that the binding of the NCK and GRB2 SH2/SH3 adaptor proteins to their cognate pTyr sites in the C-terminal cytoplasmic domain of the viral A36 protein embedded in the Vaccinia virion membrane is important for the formation of actin tails on the virion that drive intracellular virus motility and cell to cell spread. Here, they made the surprising observation that it is essential to have the NCK-binding site upstream of the GRB2-binding site for the formation of functional actin tails. This suggests that precise spatial organization of signaling protein complexes that drive actin cytoskeleton assembly is key to optimal signal output, and, by extension, this principle may be important in other signaling pathways with multiple inputs.

## Introduction

Multicellular animals extensively use phosphotyrosine (pTyr) signals for growth, communication, movement, and differentiation (*Jin and Pawson, 2012*; *Lim and Pawson, 2010*). Pathways including EGF and insulin receptor signalling, as well as T cell activation rely on pTyr motifs interacting with SH2 domains (*Blumenthal and Burkhardt, 2020*; *Lemmon and Schlessinger, 2010*). SH2 domains are present in kinases, phosphatases as well as adaptor proteins that lack enzymatic activity but couple upstream signalling events to downstream function (*Liu and Nash, 2012*). Examples of such adaptors include Shc1, Crk, Nck, and Grb2 that also contain other interaction modules such as SH3 domains that bind polyproline (PxxP) motifs (*Bywaters and Rivera, 2021*; *Mayer, 2015*). pTyr signalling is

often dysregulated in cancers and other diseases. For example, EGF receptors can be mutationally activated or present in high copy numbers vis-à-vis their cognate adaptors, and oncogenic mutations frequently map to SH2 domains (*Li et al., 2012a*; *Li et al., 2021*; *Shi et al., 2016*; *Sigismund et al., 2018*). It therefore remains important to understand the molecular principles of how pTyr-dependent signalling networks function.

pTyr motifs commonly occur in poorly ordered, unstructured regions of proteins (*Stavropoulos et al., 2012*; *Van Roey et al., 2014*). Moreover, a single polypeptide may be phosphorylated more than once to generate multiple SH2 domain binding sites creating a large network of interactions with many signalling modules. Indeed, like many receptor tyrosine kinases, the C terminus of EGFR is disordered (*Figure 1—figure supplement 1A*; *Keppel et al., 2017*; *Pinet et al., 2021*). This region contains several pTyr motifs including those that bind Grb2 and Shc1 (*Batzer et al., 1994*; *Lin et al., 2019*; *Mandiyan et al., 1996*; *Smith et al., 2006*; *Ward et al., 1996*). The relative positions of these motifs are conserved across vertebrates (*Figure 1—figure supplement 1B*). Though they are short and space-efficient, linear motifs bearing such pTyr must interact with effectors comprising globular domains of varying sizes. Additionally, disordered sequences can become structured when bound to their respective domains (*Davey, 2019*; *Nioche et al., 2002*; *van der Lee et al., 2014*). Given that such factors are likely to impose constraints or 'polarity' on the architecture of signalling networks, can the positioning of motifs have an impact on function? If yes, this would suggest that pTyr motifs cannot be repositioned as they are optimised to achieve the desired signalling output.

In recent years, the framework of phase separation or biomolecular condensates has shed new light on signalling network organisation (*Banani et al., 2017*; *Huang et al., 2019*; *Li et al., 2012b*; *Zhao and Zhang, 2020*). Integral membrane proteins with disordered cytoplasmic regions and involved in multivalent pTyr-SH2 interactions such as Linker for Activation of T cells (LAT) and the kidney podocyte regulator nephrin form phase separated condensates at critical concentrations (*Case et al., 2019*; *Ditlev et al., 2019*; *Kim et al., 2019*; *Pak et al., 2016*; *Su et al., 2016*). This striking property is thought to contribute to their signalling function. As these proteins have been largely investigated by overexpressing components in cells or by reconstitution in vitro, we are yet to fully understand how the underlying principles of condensate organisation regulate physiological signalling (*Alberti et al., 2019*; *Mayer and Yu, 2018*; *McSwiggen et al., 2019*). Furthermore, the relationship between pTyr motifs in these systems has only been explored via mutational disruption of the sites (*Huang et al., 2017*) and the role of their positioning has not been investigated. Do membrane signalling proteins make stochastic connections with their downstream components or do underlying wiring principles involving site-specific or favoured interactions exist? We chose to investigate the importance of pTyr motif positioning in a physiologically relevant model, that of Vaccinia virus egress from its infected host cell (*Leite and Way, 2015*).

Following replication and assembly, Vaccinia virus recruits kinesin-1 via WD/WE motifs in the cyto-plasmic tail of A36 an integral viral membrane protein to transport virions to the cell periphery on microtubules (*Dodding et al., 2011*; *Figure 1*). After viral fusion with the plasma membrane, extracel-lular virions that remain attached to the cell locally activate Src and Abl family kinases (*Frischknecht et al., 1999*; *Newsome et al., 2004*; *Newsome et al., 2006*; *Reeves et al., 2005*). This results in phosphorylation of tyrosine 112 and 132 in a disordered region of A36 once it incorporates into the plasma membrane when the virus fuses at the cell periphery (*Frischknecht et al., 1999*; *Newsome et al., 2004*; *Newsome et al., 2006*; *Reeves et al., 2005*; *Ward and Moss, 2004*; *Figure 1*, *Figure 1—figure supplement 1* – supplement 1C). pTyr 112 and 132 motifs bind the SH2 domains of adaptors Nck and Grb2, respectively (*Scaplehorn et al., 2002*). Nck recruits N-WASP via WIP to activate the Arp2/3 complex at the virus (*Donnelly et al., 2013*). The resulting actin polymerisation drives virus motility and enhances its cell-to-cell spread (*Ward and Moss, 2004*). Nck is essential for actin tail formation, while Grb2 recruitment helps stabilise the signalling complex (*Frischknecht et al., 1999*; *Scaplehorn et al., 2002*; *Weisswange et al., 2009*). Additionally, NPF motifs in A36 interact with the RhoGEF intersectin to recruit Cdc42 and clathrin to the virus, further enhancing actin polymerisation (*Humphries et al., 2012*; *Humphries et al., 2014*; *Snetkov et al., 2016*; *Figure 1*). The turnover rates of Nck, Grb2, and N-WASP beneath extracellular virions are highly reproducible with little variability, suggesting the signalling network has a defined organisation (*Weisswange et al., 2009*). This under-lying organisation may arise from the fact that both WIP and N-WASP only have two Nck-binding sites, each with distinct preferences for the three adaptor SH3 domains (*Donnelly et al., 2013*). This

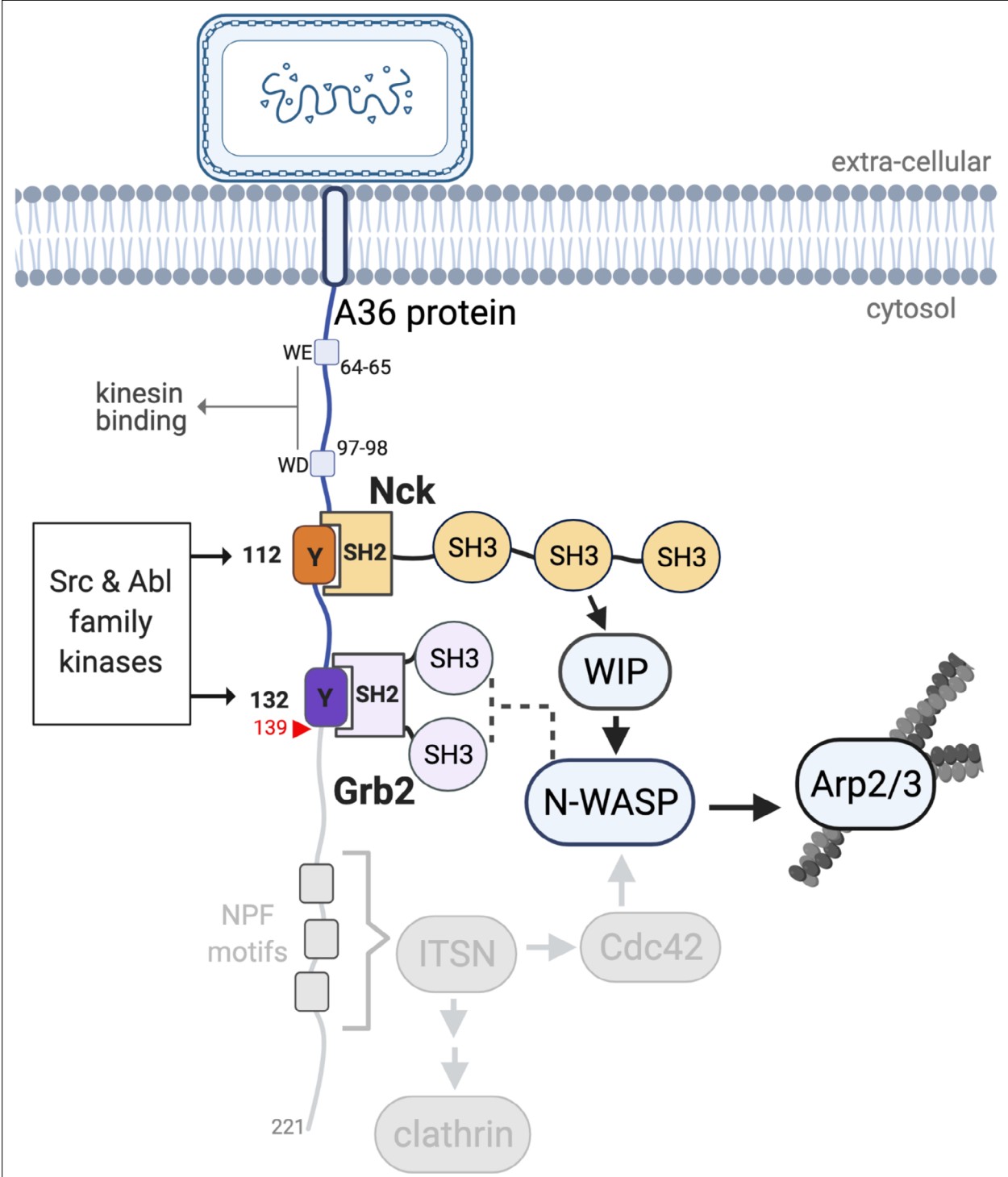

**Figure 1.** A36 interactions and the Vaccinia signalling network. A schematic showing the Vaccinia virus protein A36 and its known interactors. Kinesin-1 that drives microtubule-based transport of virions to the plasma membrane binds the WD/WE motifs. Nck and Grb2 bind Y112 and Y132 respectively when they are phosphorylated by Src and Abl family kinases. Nck and Grb2 interact with WIP and N-WASP via their SH3 domains, which results in the activation of the Arp2/3 complex and stimulation of actin polymerisation. The region deleted in A36 after residue 139 (red triangle) to abolish the involvement of the RhoGEF intersectin and its binding partners clathrin and Cdc42 is shown in grey. For simplicity, the A36 molecule has been illustrated as extending into the cytosol perpendicular to the membrane, but its exact orientation is unknown.

The online version of this article includes the following figure supplement(s) for figure 1:

**Figure supplement 1.** pTyr motifs are found in disordered regions.

Nck-dependent signalling network is not unique to Vaccinia and is also used to polymerise actin by nephrin in kidney podocytes and the pathogenic *E. coli* EPEC protein Tir (*Hayward et al., 2006*; *Jones et al., 2006*; *Welch and Way, 2013*).

Cellular pTyr networks are typically transient and can be difficult to detect, making them challenging to manipulate and study quantitatively (*Sharma et al., 2014*). In contrast, the Vaccinia virus signalling network and actin tail formation are robust and sustained. Vaccinia therefore provides an in vivo genetically tractable system where outputs of actin polymerisation, virus speed and spread can be quantitatively measured. Using recombinant Vaccinia viruses, including ones expressing Orthoreovirus protein p14 (*Figure 1—figure supplement 1C*), we uncovered a striking impairment in actin-based motility and spread of the virus when the relative positions of Nck and Grb2 pTyr binding motifs were manipulated. Our results indicate that the relative positioning of pTyr motifs and downstream adaptor binding is an important factor in the output of signalling networks that has previously been overlooked.

## Results

### A minimal system to investigate how pTyr signalling regulates actin polymerisation

The recruitment of Nck by A36 at pTyr112 is necessary and sufficient to drive actin-based motility of Vaccinia virus, whereas Grb2 recruitment at pY132 improves actin tail formation (*Frischknecht et al., 1999*; *Scaplehorn et al., 2002*; *Weisswange et al., 2009*). In addition to these two adaptors, the recruitment of intersectin, Cdc42 and clathrin by full-length A36 also indirectly regulates the extent of virus-driven actin polymerisation (*Humphries et al., 2012*; *Humphries et al., 2014*; *Snetkov et al., 2016*). To reduce the complexity arising from these interactions and other unknown A36 binding partners, we generated a recombinant virus that recruits a minimal signalling network to activate Arp2/3 complex driven actin polymerisation (*Figure 1*). To achieve this goal, the WR–ΔA36R virus lacking the A36 gene was rescued with an A36 variant that terminates after the Grb2 binding site at residue 139 (referred to as the A36 N-G virus hereafter). This shorter A36 variant retains the WD/WE kinesin-1 binding motifs that are required for microtubule-based transport of the virus to the plasma membrane. As in the full-length protein, actin tail formation induced by this shorter A36 variant depends on the Nck-binding site (*Figure 2—figure supplement 1A*, A36 N-G vs X-G). Similar to the intact protein, truncated A36 induces shorter actin tails in the absence of Grb2 recruitment (*Figure 2—figure supplement 1A*, A36 N-G vs N-X). Having confirmed the role of Nck and Grb2 is the same as in the wild-type virus, we used the A36 N-G virus to explore the role of pTyr motif positioning in signalling output.

### The relative positioning of Nck and Grb2-binding impacts actin polymerisation

Residues N-terminal to the Tyr (positions –4 to –1) are important for tyrosine kinase site recognition while SH2 domain binding specificity is typically based on +1 to +6 residues C-terminal to the pTyr (*Blasutig et al., 2008*; *Frese et al., 2006*; *Kefalas et al., 2018*; *Songyang and Cantley, 1995*; *Wagner et al., 2013*). Given these previous observations, we compared actin polymerisation, virus motility and spread of the A36 N-G virus with a recombinant virus where 12 residues surrounding the pTyr112 Nck-binding site were exchanged with the pTyr132 Grb2 interaction motif (*Figure 2A and A36* G-N virus hereafter). This modified virus maintains requirements for Src and Abl mediated phosphorylation as well as Nck and Grb2 SH2 binding. Strikingly, exchanging the positions of these two pTyr motifs impacts the length of actin tails though the number of extracellular virus particles inducing actin polymerisation remains unaltered (*Figure 2B*). It is possible that this effect is a consequence of the relative positioning of the two adaptor binding sites in A36. Alternatively, it may be that the Nck-binding site is sub-optimal when it is repositioned into the Grb2-binding locus. To determine which is true, we assessed the impact of changing the position of the Nck-binding site in the absence of Grb2 recruitment. We found that viruses A36 N-X and X-N that lack a Grb2 binding site (N-X: Nck binds in native position and X-N: Nck binds in Grb2 position), did not differ considerably in their extent or ability to induce actin tails, although there is a mild impact on actin tail length (p=0.18) (*Figure 2—figure supplement 1B*). Curiously, having two Nck or Grb2 binding sites does not improve

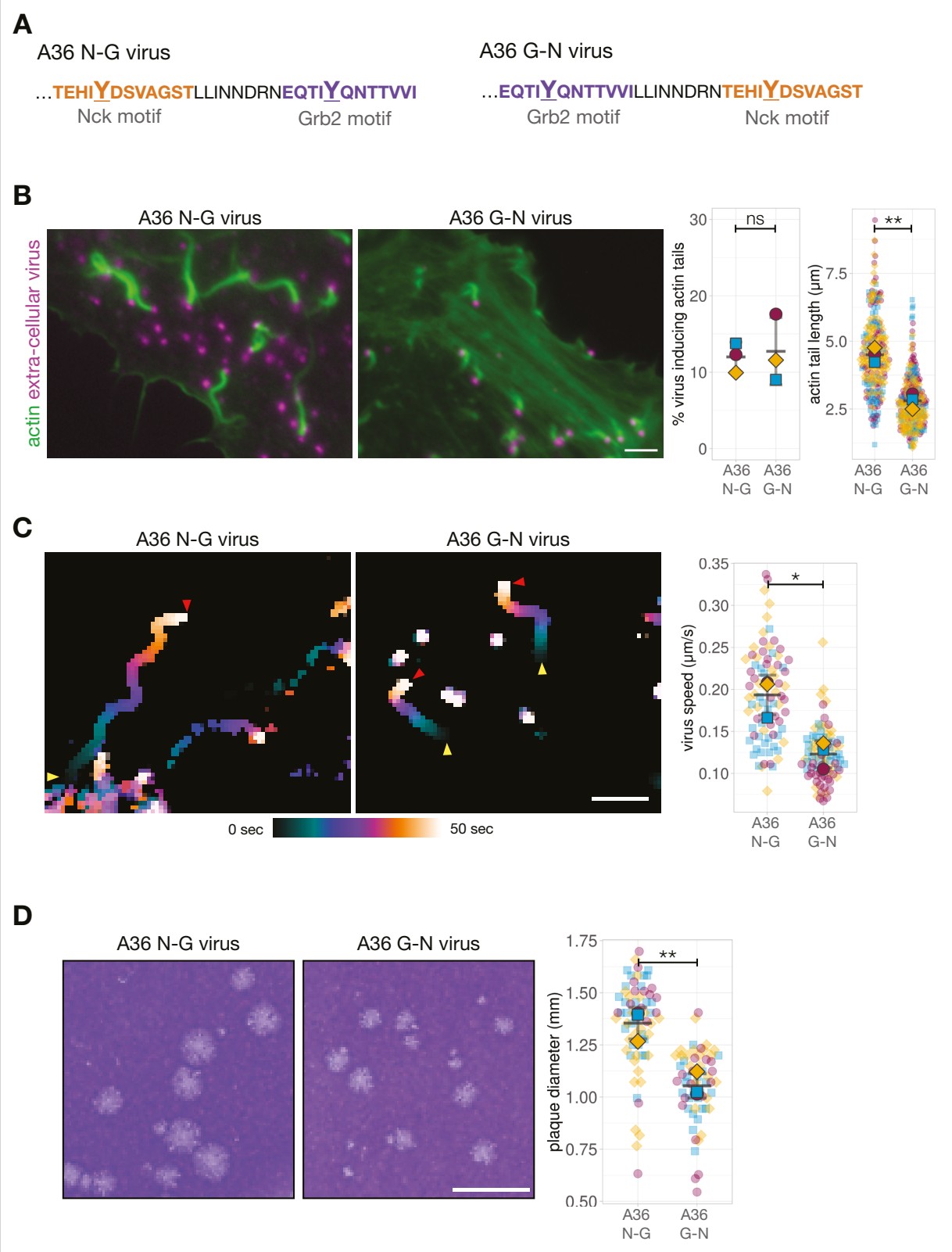

**Figure 2.** Phosphotyrosine motif position impacts actin-based motility and viral spread. (**A**) C-terminal amino acid sequence of A36 in recombinant viruses showing the position of phosphotyrosine motifs in their wild-type (A36 N-G) and swapped (A36 G-N) configurations. (**B**) Representative immunofluorescence images of actin tails in HeLa cells infected with the indicated virus at 8 hr post-infection. Actin is stained with phalloidin, and extra-cellular virus particles attached to plasma membrane are labelled using an antibody against the viral protein B5. Scale bar = 3 μm. The graphs

*Figure 2 continued on next page*

*Figure 2 continued*

show quantification of number of extracellular virus particles inducing actin tails and their length. A total of 270 actin tails were measured in three independent experiments. (**C**) Temporal colour-coded representation of time-lapse movies tracking the motility of the indicated RFP-A3-labelled virus over 50 s at 8 hr post infection (***Video 1***). Images were recorded every second and the position of virus particles at frame 1 (yellow triangles) and frame 50 (red triangles) are indicated. Scale bar = 3 μm. The graph shows quantification of virus speed over 50 s. A total of 82 virus particles were tracked in three independent experiments. (**D**) Representative images and quantification of plaque diameter produced by the indicated virus in confluent BS-C-1 cells 72 hr post-infection. Sixty-four plaques were measured in three independent experiments. Scale bar = 3 mm. All error bars represent S.D and the distribution of the data from each experiment is shown using a 'SuperPlot'. Welch's t test was used to determine statistical significance; ns, p>0.05; * p≤0.05; ** p≤0.01.

The online version of this article includes the following source data and figure supplement(s) for figure 2:

**Source data 1.** Datasheets for graphs and summary statistics.

**Figure supplement 1.** Quantification of the number of virus inducing actin tails along with their length and characterisation of A36 G-N recombinant viruses.

**Figure supplement 1—source data 1.** Datasheets for graphs and summary statistics.

**Figure supplement 2.** Quantification of the number of virus inducing actin tails along with their length.

**Figure supplement 2—source data 1.** Datasheets for graphs and summary statistics.

the ability of a single adaptor to promote actin polymerisation (***Figure 2—figure supplement 2A***). The length of actin tails was also not influenced by extending the linker between the adaptor binding sites (***Figure 2—figure supplement 2B***). Collectively, our observations suggest that the relative positioning and number of Nck and Grb2 phosphotyrosine motifs in A36 is optimised for the signalling output (efficient actin polymerisation).

Actin polymerisation at the virus drives both the motility and spread of virions. To test whether motif positioning influences the former, we imaged cells infected with the A36 N-G or A36 G-N viruses expressing A3, a viral core protein tagged with RFP. In the swapped configuration, corresponding to reduced actin polymerisation, virus motility is slower (***Figure 2C***, ***Video 1***). In addition, the A36 G-N virus has reduced cell-to-cell spread as it forms smaller plaques on confluent cell monolayers compared to A36 N-G (***Figure 2D***; ***Figure 2—figure supplement 1C***). Our observations with Vaccinia clearly demonstrate that the output of a signalling network is strongly influenced by the relative positioning of Nck and Grb2 binding sites in the membrane protein responsible for initiating the signalling cascade.

## Inducing actin polymerisation using a synthetic signalling network

We were curious whether our observations were unique to Vaccinia A36 and/or if the positioning of adaptor binding sites would also be important in a different context. We took advantage of the Vaccinia signalling platform to generate a synthetic pTyr network that replaces A36 with a different protein capable of activating actin polymerisation via the same adaptors. Unfortunately, we could not identify a host receptor that signals to actin via Nck and Grb2. Recent observations, however, demonstrate that p14, an integral membrane protein from Orthoreovirus activates N-WASP via Grb2 to regulate cell fusion (***Chan et al., 2020***). Grb2 is recruited by a pTyr116 motif located in the short cytoplasmic tail of p14 that is also predicted to be disordered (***Figure 1—figure supplement 1C***). Interestingly, examination of the p14 sequence reveals there are two tyrosine residues predicted to interact with Nck located 16 and 20 amino acids upstream of the pTyr116 (Eukaryotic Linear Motif (ELM) and Scansite 4.0) (***Kumar et al., 2020***; ***Obenauer et al., 2003***; ***Figure 3A***). To determine whether these sites participate in actin polymerisation, we generated a hybrid construct comprising the first 105

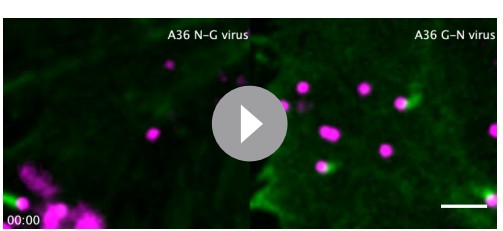

**Video 1.** Phosphotyrosine motif position impacts actin-based motility of Vaccinia virus. HeLa cells stably expressing LifeAct-iRFP670 (green) were infected with either the A36 N-G or A36 G-N virus labelled with RFP-A3 (magenta) for 8 hr. Images were taken every second. Video plays at 5 frames per second. The RFP-A3 signal was used to generate the temporal colour-coded representation in Figure 2C. The time in seconds is indicated, and the scale bar = 3 μm.
https://elifesciences.org/articles/74655/figures#video1

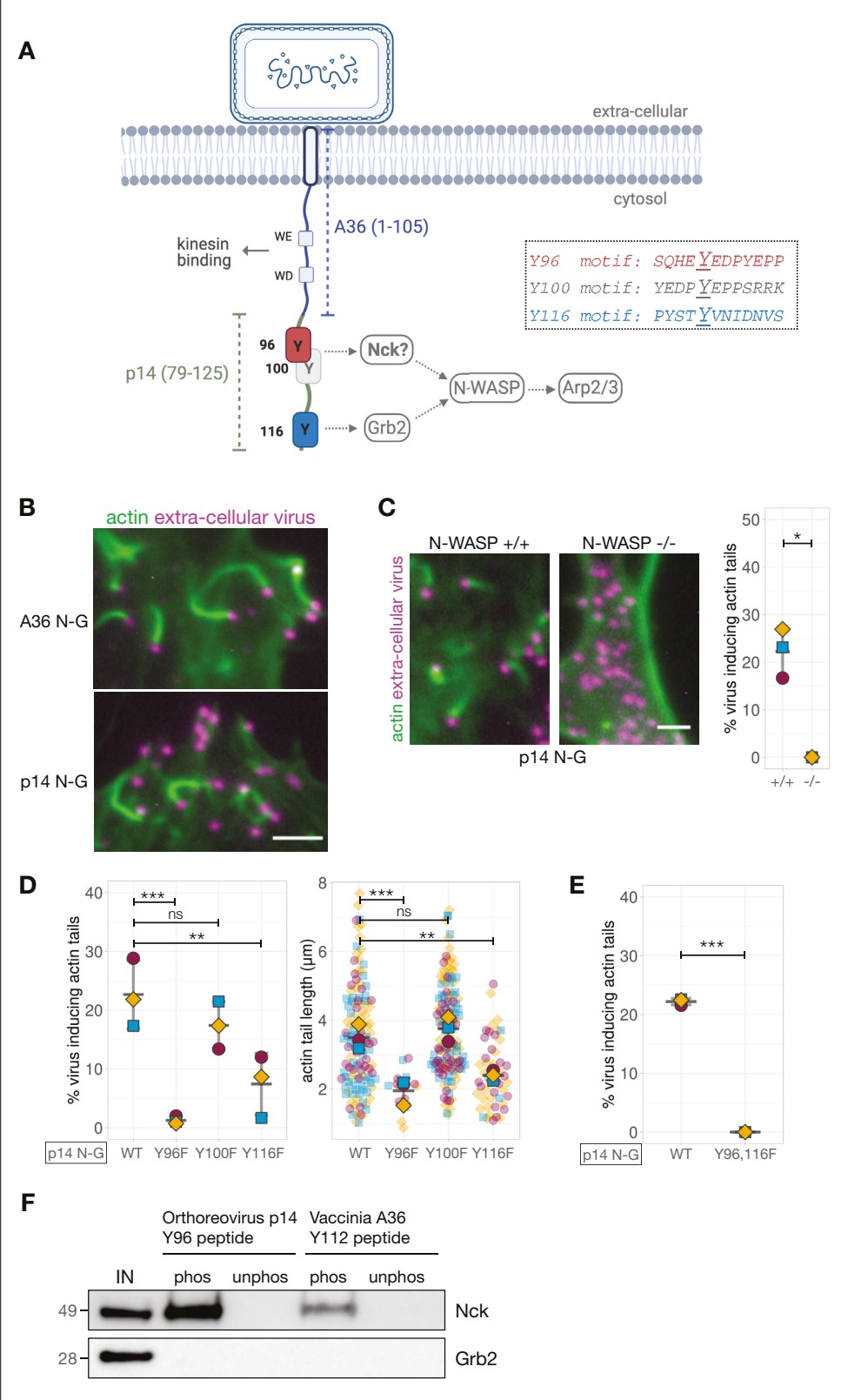

**Figure 3.** Generating and validating an A36-p14 hybrid that can polymerise actin. (**A**) Schematic showing a hybrid construct (referred to as p14 N-G) comprising the first 105 residues of A36 and the C-terminal residues 79–125 of Orthoreovirus p14 protein. Positions of the predicted Nck-binding sites at Tyr96 and Tyr100, and the previously established Grb2-binding site at Tyr116 are shown together with their respective sequences. (**B**) Representative

*Figure 3 continued on next page*

*Figure 3 continued*

immunofluorescence images of actin tails in HeLa cells infected with Vaccinia virus lacking the A36 gene and transiently expressing the indicated constructs under the A36 promoter at 8 hr post-infection. Actin is stained with phalloidin, and extra-cellular virus particles attached to plasma membrane are labelled using an anti-B5 antibody. Scale bar = 3 µm. (**C**) Representative immunofluorescence images of N-WASP null or parental mouse embryonic fibroblast cells infected with Vaccinia virus lacking the A36 gene and transiently expressing the p14 N-G construct under the A36 promoter at 16 hr post-infection. Actin is stained with phalloidin, and extra-cellular virus particles are labelled using an anti-B5 antibody. Scale bar = 3 µm. The graph shows quantification of actin tail number per extracellular virus particle. Error bars represent S.D. from three independent experiments. Welch's t test was used to determine statistical significance; * p≤0.05. (**D**) and (**E**) Quantification of the number of extracellular virus inducing actin tails together with their length in HeLa cells infected with Vaccinia virus lacking the A36 gene and transiently expressing p14 N-G constructs under the A36 promoter with indicated Tyr to Phe mutations, at 8 hr post-infection. 125 actin tails were measured in three independent experiments, except in mutants Y96F and Y116F where fewer actin tails were made. All error bars represent S.D and the distribution of the data from each experiment is shown using a 'SuperPlot'. Dunnett's multiple comparison's test (for panel D) and Welch's t test (for panel E) were used to determine statistical significance; ns, p>0.05; ** p≤0.01; *** p≤0.001. (**F**) Immunoblot analysis of peptide pulldowns showing that endogenous Nck from HeLa cell lysates binds to phosphopeptides corresponding to Tyr96 from the Orthoreovirus p14 and Tyr112 of the Vaccinia A36 but not to their unphosphorylated counterparts.

The online version of this article includes the following source data and figure supplement(s) for figure 3:

**Source data 1.** Datasheets for graphs, summary statistics and raw immunoblots.

**Figure supplement 1.** Generating and validating an A36-p14 hybrid that can polymerise actin.

**Figure supplement 1—source data 1.** Raw immunoblots.

amino acids of A36 including the kinesin-1-binding motifs required to traffic virions to the plasma membrane and residues 79–125 of p14 (*Figure 3A*). Transient expression of p14 N-G in cells infected with Vaccinia virus lacking A36 results in extracellular virions inducing robust actin tails that were dependent on the presence of N-WASP (*Figure 3B and C*). Given this, we mutated Tyr96, 100 and 116 of p14 in turn and examined the impact on actin tail formation. This analysis reveals there is no role for Tyr100 as its mutation to phenylalanine had no impact on actin tail formation (*Figure 3D*). In contrast, Tyr96 is essential for the virus induced actin polymerisation. As with Vaccinia A36, the ability of transiently expressed p14 N-G to promote actin polymerisation primarily depends on the predicted Nck-binding pTyr96 and is enhanced by pTyr116 (*Figure 3D and E*). Disrupting these sites via Y96F and Y116F mutations in p14 N-G results in loss of Nck and Grb2 recruitment to the virus respectively (*Figure 3—figure supplement 1A*). When phosphorylated, Tyr116 is a bona fide Grb2-binding site (*Chan et al., 2020*). To verify pTyr96 interacts with Nck, we incubated HeLa cell lysates with beads conjugated with peptides containing the prospective binding motif. The pTyr96 peptide retains Nck from the cell lysate but not the unphosphorylated control (*Figure 3F*). The peptide also does not bind Grb2. Consistent with our observation that Tyr100 plays no part in actin polymerisation, the pTyr100 peptide does not bind either adaptor (*Figure 3—figure supplement 1B*). Our observations clearly demonstrate that as with Vaccinia A36, p14 of Orthoreovirus induces actin polymerisation using a Nck and Grb2 signalling network. These results also establish that the Vaccinia platform can be used to establish synthetic signalling networks and test predictions concerning their operation.

Given this, we generated a recombinant virus where endogenous A36 was replaced with the hybrid A36-p14 protein (henceforth referred to as the p14 N-G virus). As previously observed with Vaccinia A36, actin tails generated by the p14 N-G virus recruit endogenous Nck, WIP, and N-WASP to the virion (*Figure 4A*). In the absence of an antibody that can detect Grb2, we confirmed the p14 N-G virus also recruits Grb2 by infecting HeLa cells stably expressing GFP-Grb2 (*Figure 4A*; *Figure 4—figure supplement 1*). We next sought to determine whether the relative positioning of the Nck and Grb2 binding sites was important for the signalling output of p14. To address this question, we generated a recombinant virus (p14 G-N virus) where the Nck- and Grb2-binding motifs of p14 are in a swapped orientation (*Figure 4B*). Remarkably, as we observed for A36, swapping these motifs did not impact on the number of viruses inducing actin polymerisation but resulted in shorter actin tails and a slower virus speed (*Figure 4C and D*, *Video 2*). Plaque assays on confluent cell monolayers reveals that the p14 G-N virus is also attenuated in its spread when compared to p14 N-G (*Figure 4E*). Once

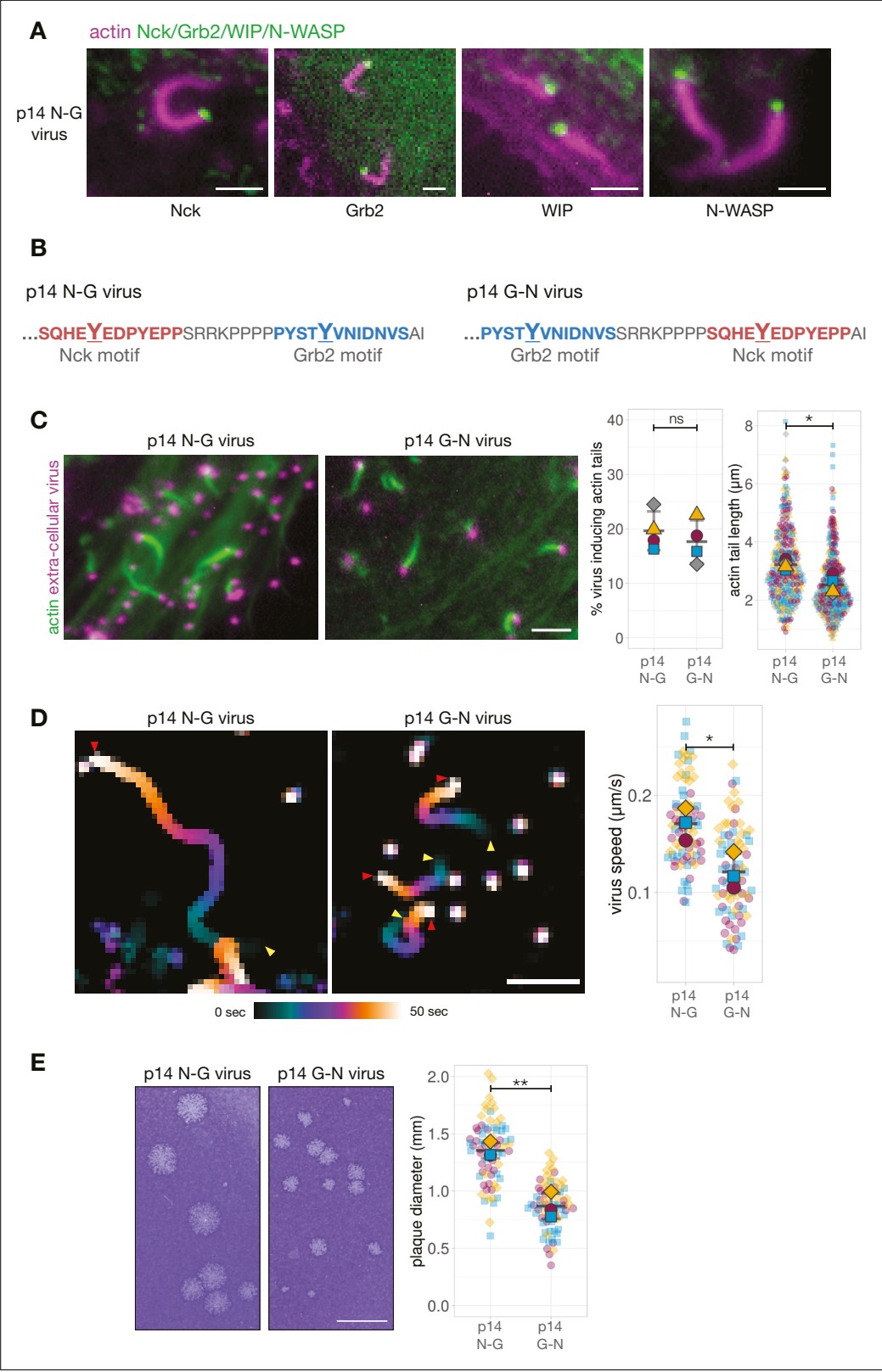

**Figure 4.** Phosphotyrosine motif position in a A36-p14 hybrid protein impacts actin polymerisation.
(**A**) Representative images showing the recruitment of Nck, WIP, Grb2, and N-WASP to actin tails in HeLa cells infected with a recombinant virus expressing the p14 N-G construct at the A36 locus at 8 hr post-infection. Endogenous Nck, N-WASP and WIP were detected with antibodies and actin is labelled with phalloidin. To

*Figure 4 continued on next page*

*Figure 4 continued*

ascertain Grb2 localisation, infected HeLa cells stably expressing GFP-Grb2 and transiently expressing LifeAct iRFP were imaged live. Scale bar = 2 µm. (**B**) C-terminal amino acid sequence of the A36 -p14 hybrid in recombinant viruses showing the positions of phosphotyrosine motifs in their wild-type (p14 N-G) and swapped (p14 G-N) configurations. (**C**) Representative immunofluorescence images of actin tails in HeLa cells infected with the indicated virus at 8 hr post-infection. Actin is stained with phalloidin, and extra-cellular virus particles are using an anti-B5 antibody. Scale bar = 3 µm. The graphs show quantification of number of extracellular virus particles inducing actin tails and their length. 336 actin tails were measured in four independent experiments. (**D**) Temporal colour-coded representation of time-lapse movies tracking the motility of the indicated RFP-A3-labelled virus over 50 s at 8 hr post infection (*Video 2*). Images were recorded every second and the position of virus particles at frame 1 (yellow triangles) and frame 50 (red triangles) are indicated. Scale bar = 3 µm. The graph shows quantification of virus speed over 50 s. Seventy-five virus particles were tracked in three independent experiments. (**E**) Representative images and quantification of plaque diameter produced by the indicated virus in confluent BS-C-1 cells 72 hr post-infection. Seventy-two plaques were measured in three independent experiments. Scale bar = 3 mm. All error bars represent S.D and the distribution of the data from each experiment is shown using a 'SuperPlot'. Welch's t test was used to determine statistical significance; ns, p>0.05; * p≤0.05; ** p≤0.01.

The online version of this article includes the following source data and figure supplement(s) for figure 4:

**Source data 1.** Datasheets for graphs and summary statistics.

**Figure supplement 1.** Validation of stable cell lines by immunoblot.

**Figure supplement 1—source data 1.** Raw immunoblots.

---

again, the signalling output of the system clearly depends on the relative positioning of Nck and Grb2 pTyr binding sites.

## Why does the relative position of adaptor binding sites impact actin polymerisation?

To further probe into how pTyr motif positioning impacts signalling output, we focused on the A36 signalling network given we have a deeper understanding of how it functions. The simplest explanation for the difference between the signalling output of the N-G and G-N configurations is that the levels of A36 are different between the two viruses. To examine if this is the case, we generated recombinant viruses where the A36 N-G and G-N variants were tagged at their C-terminus with TagGFP2. We measured the fluorescent intensity ratio of TagGFP2 to the viral core protein A3 fused to RFP. RFP-A3 provides a reliable internal reference marker as its fluorescent intensity is consistent across different viruses and cell lines expressing GFP-tagged components of the Vaccinia signalling network (*Figure 5—figure supplement 1A*). We found no significant difference between the levels of A36 N-G and G-N relative to RFP-A3 on particles generating actin tails (*Figure 5—figure supplement 1B*). This suggests that the underlying cause for the difference in signalling output resides in the network itself. We therefore determined the level of recruitment of components involved in activating the Arp2/3 complex on the virus. We infected HeLa cells stably expressing GFP-tagged Nck, Grb2, WIP, or N-WASP (*Figure 4—figure supplement 1*) and measured their respective GFP intensities on the virus relative to RFP-A3 as an internal fluorescent standard. We found that the levels of Nck are comparable between the A36 N-G and G-N viruses (*Figure 5A*). In contrast, Grb2 and N-WASP were significantly reduced on the G-N virus (*Figure 5A*). The level of WIP, which is also recruited to actin tails is lower in the G-N configuration (*Figure 5—figure supplement 1C*). To confirm these results, we took advantage of Mouse Embryonic Fibroblast (MEF) cell lines stably expressing GFP-tagged Nck- or N-WASP

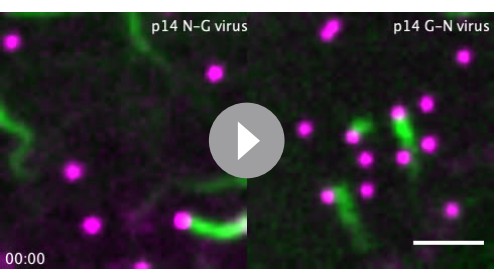

**Video 2.** Phosphotyrosine motif position impacts actin-based motility driven by p14 N-G virus. HeLa cells stably expressing LifeAct-iRFP670 (green) were infected with either the p14 N-G or p14 G-N virus labelled with RFP-A3 (magenta) for 8 hr. Images were taken every second. Video plays at 5 frames per second. The RFP-A3 signal was used to generate the temporal colour-coded representation in Figure 4D. The time in seconds is indicated, and the scale bar = 3 µm.

https://elifesciences.org/articles/74655/figures#video2

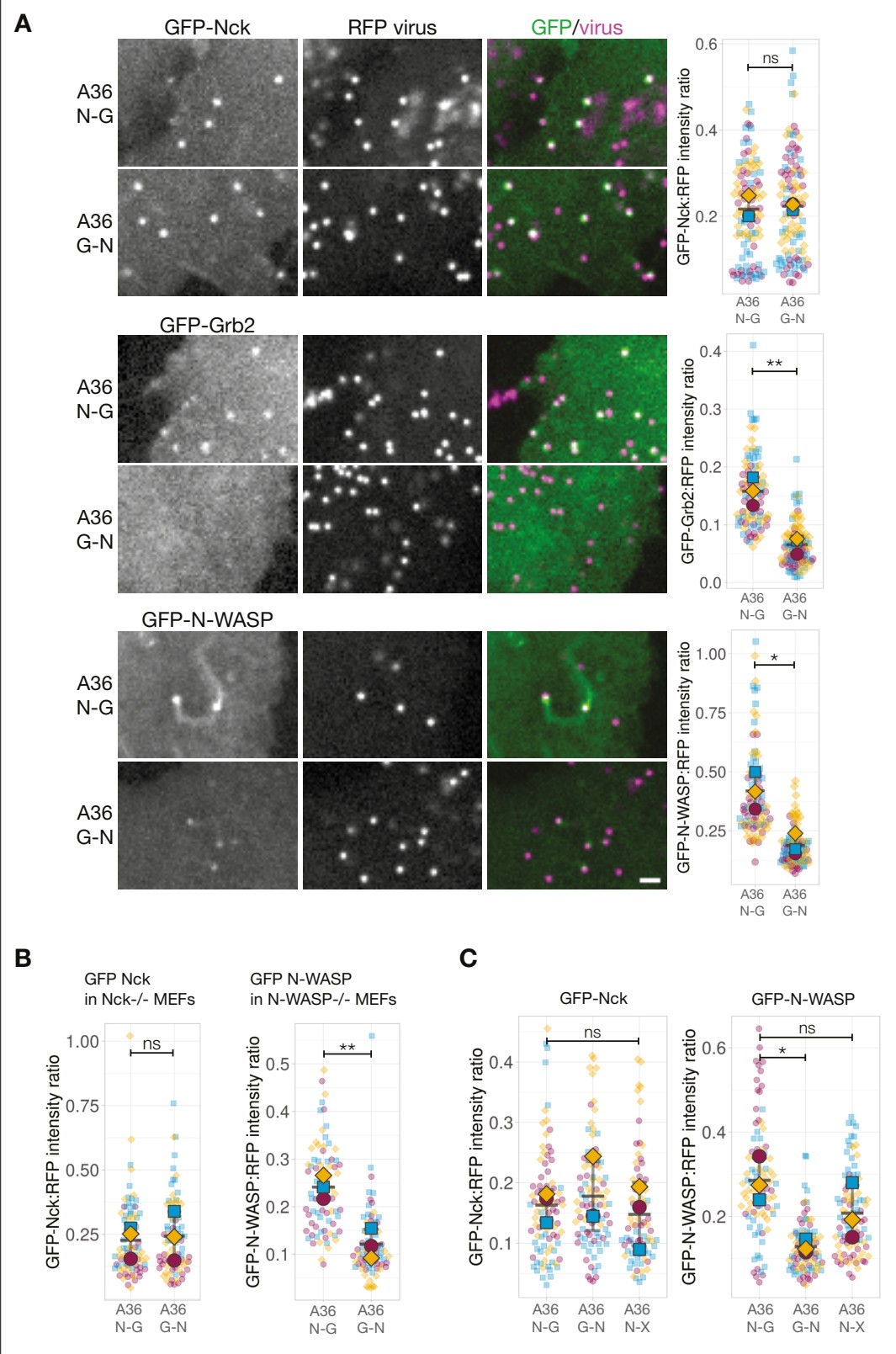

**Figure 5.** N-WASP recruitment is impaired when Grb2 binding is repositioned in A36. (**A**) Representative images showing the indicated GFP-tagged protein recruitment to RFP-A3 labelled virus particles in live HeLa cells infected with the indicated viruses at 8 hr post-infection. Scale bar = 2 µm. The graphs show quantification of GFP:RFP-A3 fluorescence intensity ratio. Intensity of 90 virus particles was measured in three independent experiments.

*Figure 5 continued on next page*

*Figure 5 continued*

(**B**) Left - The graph shows quantification of GFP-Nck:RFP-A3 fluorescence intensity ratio on virus particles in live mouse embryonic fibroblasts (MEFs) lacking both Nck1 and Nck2 and stably expressing GFP-Nck1 at 16 hr post-infection. Intensity of 75 virus particles was measured in three independent experiments. Right - The graph shows quantification of GFP-N-WASP:RFP-A3 fluorescence intensity ratio on virus particles in live N-WASP-/- MEFs stably expressing GFP-N-WASP infected with the indicated viruses at 16 hr post-infection. Intensity of 75 virus particles was measured in three independent experiments. (**C**) The graphs show quantification of GFP:RFP-A3 fluorescence intensity ratio on virus particles in live HeLa cells stably expressing the indicated GFP-tagged protein infected with the A36 N-G, G-N, or N-X viruses at 8 hr post-infection. Intensity of 90 virus particles was measured in three independent experiments. All error bars represent S.D and the distribution of the data from each experiment is shown using a 'SuperPlot'. Dunnett's multiple comparison's test (for panel C) or Welch's t test (remaining panels) were used to determine statistical significance; ns, p>0.05; * p≤0.05; ** p≤0.01.

The online version of this article includes the following source data and figure supplement(s) for figure 5:

**Source data 1.** Datasheets for graphs and summary statistics.

**Figure supplement 1.** Fluorescence intensity measurements of Vaccinia recombinants.

**Figure supplement 1—source data 1.** Datasheets for graphs and summary statistics.

**Figure supplement 2.** Verification of stable cell lines by immunoblot.

**Figure supplement 2—source data 1.** Raw immunoblots.

but lacking their respective endogenous proteins (*Figure 5—figure supplement 2*). We found that as seen in HeLa cells, the levels of GFP-Nck were similar for both viruses but the A36 G-N virus recruited twofold less GFP-N-WASP than the N-G variant (*Figure 5B*). It is possible that the reduction in N-WASP is a consequence of lower Grb2 levels as this adaptor helps stabilise N-WASP on the virus (*Weisswange et al., 2009*). To examine this possibility, we measured N-WASP levels on the A36 N-X virus where the Grb2-binding site is abolished by a Tyr132 to Phe mutation. Loss of Grb2 binding had no appreciable impact on the levels of Nck recruited to the A36 N-X virus as compared to those associated with the A36 N-G and G-N viruses (*Figure 5C*). The levels of N-WASP on the A36 N-X virus were reduced (46.4%) but not to the same extent as the A36 G-N virus (75%). Repositioning the Nck and Grb2 sites clearly leads to a more severe impairment of the ability of the signalling network to activate Arp2/3 complex driven actin polymerisation than loss of the Grb2 site. Moreover, this suggests that Grb2 has a dominant negative effect when it is mis-positioned relative to Nck.

## Adaptor motif repositioning does not affect tyrosine phosphorylation of A36

A possible explanation for the impairment in the A36 G-N signalling network is that adaptor motif repositioning impacts kinase recruitment and/or its ability to induce tyrosine phosphorylation. Immunoblot analysis of whole cell lysates with anti-phosphotyrosine antibody demonstrates that A36 phosphorylation in the A36 G-N virus infected cells is comparable to that seen with the A36 N-G virus (*Figure 6—figure supplement 1A*). This bulk assay reports on the total cellular pool of A36 rather than the specific fraction of A36 that is phosphorylated by Src and Abl family kinases at the plasma membrane (*Newsome et al., 2004*). To directly investigate whether kinase recruitment to individual virus particles is affected by motif position, we examined the recruitment of Src-GFP to RFP-A3 labeled A36 N-G or A36 G-N viruses in HeLa cells stably expressing LifeAct iRFP. Quantification of Src-GFP at the tip of actin tails reveals that the A36 N-G and G-N viruses recruit similar levels of the tyrosine kinase (*Figure 6A*). Furthermore, immunofluorescence analysis with a Src antibody that detects phosphorylated tyrosine 418 (Src pY418) reveals a similar level of activated kinase is associated with both viruses (*Figure 6B*).

The comparable levels of Nck recruitment to the A36 N-G and G-N viruses (*Figure 5A*) suggests that phosphorylation of Tyr112 (the Nck-binding site) is not impacted by changing its position within A36. To confirm that this is the case in the absence of downstream components, we examined the ability of A36 N-G and G-N viruses to recruit GFP-Nck SH2 in MEFs lacking both Nck1 and Nck2 (*Bladt et al., 2003*). GFP-Nck SH2 is a specific probe for pTyr112 as it is not recruited to the A36 X-G virus which lacks this site (*Figure 6—figure supplement 1B*). We found that both A36 N-G and G-N viruses were equally efficient at recruiting GFP-Nck SH2 (*Figure 6C*). These data clearly demonstrate

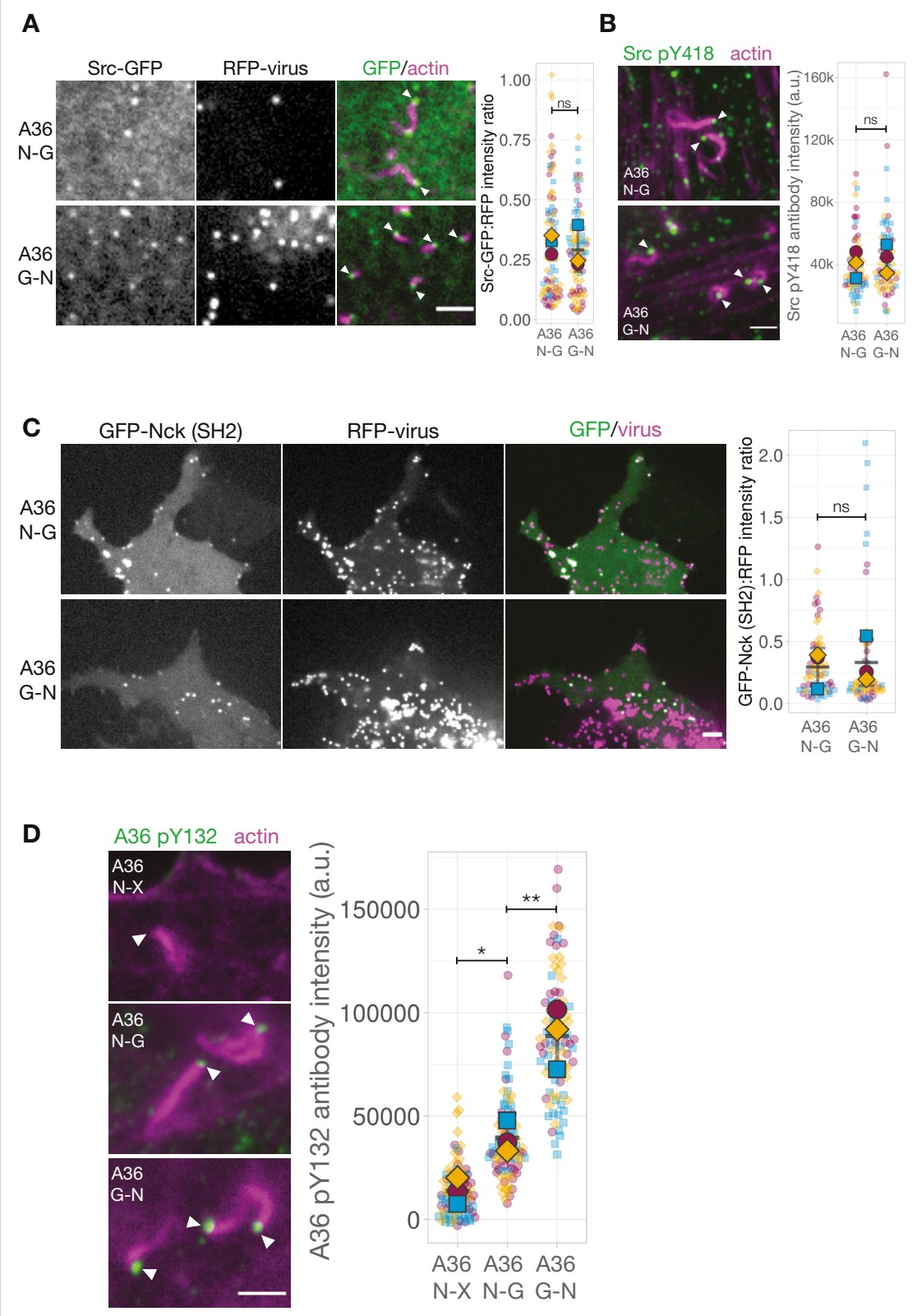

**Figure 6.** The Nck and Grb2-binding sites in A36 are phosphorylated when motif positions are changed. (**A**) Representative images showing Src-GFP recruitment to RFP-A3 labelled virus particles in live HeLa cells infected with the indicated viruses at 8 hr post-infection. Scale bar = 4 μm. The graphs show quantification of GFP:RFP-A3 fluorescence intensity ratio. Intensity of 90 virus particles was measured in three independent experiments. (**B**) Representative immunofluorescence images of Src pY418 antibody labelling of indicated virus inducing actin tails in HeLa cells at 8 hr post-infection.

*Figure 6 continued on next page*

*Figure 6 continued*

Actin is stained with phalloidin. Scale bar = 2 µm. The graph shows quantification of background-subtracted antibody intensity at the tip of actin tails (k=1000 a.u.). Intensity of 75 virus particles was measured in three independent experiments.(C) Representative images showing GFP-Nck (SH2) recruitment to RFP-A3-labelled virus particles in live Nck-/- MEF cells infected with the indicated viruses at 16 hr post-infection. Scale bar = 2 µm. The graphs show quantification of GFP:RFP-A3 fluorescence intensity ratio. Intensity of 75 virus particles was measured in three independent experiments. (**D**) Representative immunofluorescence images of A36 pY132 antibody labelling of indicated virus inducing actin tails in HeLa cells at 8 hr post-infection. Actin is stained with phalloidin. A36 N-X is a virus where the A36 Grb2-binding site is disrupted with by Tyr to Phe point mutation. Scale bar = 2 µm. The graph shows quantification of background-subtracted antibody intensity at the tip of actin tails. Intensity of 90 virus particles was measured in three independent experiments. All error bars represent S.D. and the distribution of the data from each experiment is shown using a 'SuperPlot'. Tukey's multiple comparison's test (for panel D) and Welch's t test (remaining panels) were used to determine statistical significance; ns, p>0.05; * p≤0.05; ** p≤0.01. White arrowheads indicate virus position.

The online version of this article includes the following source data and figure supplement(s) for figure 6:

**Source data 1.** Datasheets for graphs and summary statistics.

**Figure supplement 1.** The Nck and Grb2-binding sites in A36 are phosphorylated when motif positions are changed.

**Figure supplement 1—source data 1.** Datasheets for graphs, summary statistics, and raw immunoblots.

that phosphorylation of the Nck binding motif is not affected by its position. As Grb2 recruitment is impaired in the G-N virus we also sought to determine whether phosphorylation of its binding site (Tyr132) is reduced. Immunofluorescence analysis with an antibody that detects pTyr132 (*Newsome et al., 2004*) reveals that tyrosine 132 is phosphorylated on the A36 N-G and G-N but not N-X (negative control lacking Grb2 binding site) virus particles inducing actin tails (*Figure 6D*). The lack of labelling of the N-X virus demonstrates that the antibody is specific as it does not detect any other phosphorylated component in the signalling network involved in nucleating actin tails. Moreover, the intensity of labelling was significantly higher on the G-N compared to the N-G virus, suggesting there is no impairment in phosphorylation. This enhanced labelling intensity most likely reflects increased antibody access to its pTyr132 epitope in the absence of Grb2 recruitment (*Figure 5A*). To test this possibility, we determined the level of pTyr132 antibody labelling on the A36 N-G virus when Grb2 is knocked down using siRNA. We found that there was a trend for increased labelling on the A36 N-G virus when the levels of Grb2 were reduced (*Figure 6—figure supplement 1C*). This increase is less dramatic than the G-N virus most likely because the knockdown is not complete and the virus efficiently recruits any remaining Grb2. Taken together, our observations suggest that the lower levels of Grb2 recruitment observed when its binding site is repositioned are a consequence of other changes in the signalling network rather than a reduction in phosphorylation of Tyr132.

## Position rather than the number of Grb2 sites influences actin polymerisation

Our data indicate that the reduced signalling output of the A36 G-N virus is linked to decreased levels of Grb2 in the signalling network. This suggests that the position of Grb2 binding relative to Nck is the critical factor influencing the functional output of the network formed by the A36 G-N virus. We therefore wondered what impact an additional Grb2 site would have on virus-induced actin polymerisation? More importantly, would any improvement in signalling output be dependent on the position of this second Grb2 site relative to Nck binding? To address these questions, we generated recombinant G-N viruses expressing A36 with an extra Grb2 site N-terminal (A36 G-G-N virus) or C-terminal (A36 G-N-G virus) to the Nck site (*Figure 7A*). The spacing and linkers between all new motifs were the same (20 amino acids) as in the original A36 N-G virus. All four viruses (N-G, G-N, G-N-G, and G-G-N) recruited similar levels of Nck and induced comparable numbers of actin tails (*Figure 7—figure supplement 1A* and B). Nevertheless, the A36 G-N and G-G-N viruses induced the formation of equally short actin tails (*Figure 7B*). In contrast, the actin tails formed by the A36 G-N-G virus are noticeably longer. A similar trend was also observed for virus speed and spread (*Figure 7C and D*; *Figure 7—figure supplement 2*). Strikingly, the G-N-G and N-G viruses recruit similar levels of N-WASP, which are ~2 fold greater than the G-G-N and G-N viruses (*Figure 7E*). They also recruit significantly more GFP-Grb2 (*Figure 7—figure supplement 1C*). Grb2 is not essential for Vaccinia actin tail formation but its binding position relative to Nck clearly influences the level of N-WASP recruitment. This subsequently influences the output of the signalling network stimulated by phosphorylation of A36.

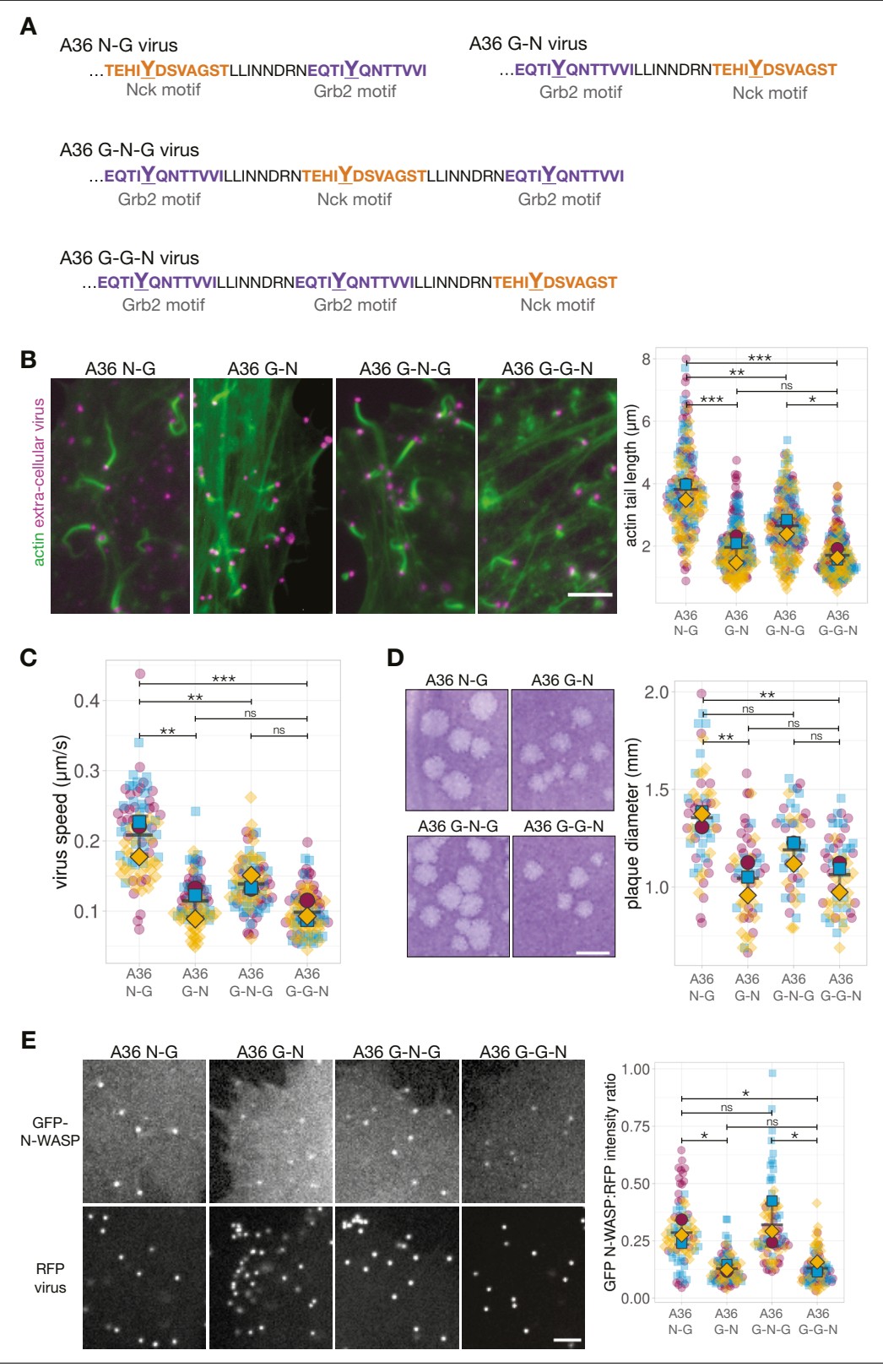

**Figure 7.** Signalling output can be improved by adding a new Grb2 binding site, but only in a position-dependent fashion. (**A**) C-terminal amino acid sequence of A36 in recombinant viruses showing the position of phosphotyrosine motifs in wild-type (A36 N-G) and swapped (A36 G-N) configurations and with a new Grb2-binding site C-terminal (**G–N–G**) or N-terminal (**G–G–N**) to the swapped configuration. (**B**) Representative

*Figure 7 continued on next page*

*Figure 7 continued*

immunofluorescence images of actin tails in HeLa cells infected with the indicated virus at 8 hr post-infection. Actin is stained with phalloidin, and extra-cellular virus particles are labelled using an anti-B5 antibody. Scale bar = 5 μm. The graph shows quantification of actin tail length. 195 actin tails were measured in three independent experiments. (**C**) The graph shows quantification of virus speed from time-lapse movies tracking the motility of the indicated RFP-A3-labelled virus over 50 s at 8 hr post infection. Ninety virus particles were tracked in three independent experiments. (**D**) Representative images and quantification of plaque diameter produced by the indicated virus in confluent BS-C-1 cells 72 hr post-infection. Sixty plaques were measured in three independent experiments. Scale bar = 2 mm. (**E**) Representative images showing GFP-N-WASP and RFP-A3 intensity on virus particles in live HeLa cells infected with the indicated viruses recorded 8 hr post-infection. Scale bar = 4 μm. GFP N-WASP intensity data for A36 N-G and G-N viruses is the same as in *Figure 5C*. The graph shows quantification of GFP-N-WASP:RFP-A3 fluorescence intensity ratio. Intensity of 90 virus particles was measured in three independent experiments. All error bars represent S.D and the distribution of the data from each experiment is shown using a 'SuperPlot'. Tukey's multiple comparison test was used to determine statistical significance; ns, p>0.05; * p≤0.05; ** p≤0.01; *** p≤0.001.

The online version of this article includes the following source data and figure supplement(s) for figure 7:

**Source data 1.** Datasheets for graphs and summary statistics.

**Figure supplement 1.** Quantification of the number of virus inducing actin tails and GFP-Nck recruitment in A36 variants with an extra Grb2-binding site.

**Figure supplement 1—source data 1.** Datasheets for graphs and summary statistics.

**Figure supplement 2.** Characterisation of A36 G-N-G and A36 G-G-N recombinant viruses.

**Figure supplement 2—source data 1.** Datasheets for graphs and summary statistics.

## Discussion
## Grb2 modulates the ability of Nck to promote actin polymerisation via N-WASP

The pTyr motifs in Vaccinia A36 engage the SH2-SH3 adaptors Nck and Grb2, which play essential roles in many different cellular signalling cascades. For example, Grb2 directly binds EGFR and Ras via its SH2 domain, while the SH2 domain of Nck binds PDGFR and ephrinb1 receptors among others (*Bong et al., 2004*; *Cowan and Henkemeyer, 2018*; *Lettau et al., 2009*; *Nishimura et al., 1993*; *Pramatarova et al., 2003*). These adaptors also participate in signalling networks that can undergo phase transitions, including those assembled by the membrane proteins LAT and nephrin in T-cell activation and kidney podocytes, respectively (*Case et al., 2019*; *Ditlev et al., 2019*; *Kim et al., 2019*; *Pak et al., 2016*; *Su et al., 2016*). LAT engages Grb2 for its function while nephrin relies on Nck. In the case of Vaccinia, Nck and Grb2 work together in a signalling network that induces Arp2/3 complex-dependent actin polymerisation to enhance the spread of viral infection (*Frischknecht et al., 1999*; *Ward and Moss, 2004*).

The virus provides a great model system to dissect this signalling network as it can be easily manipulated using recombinant viruses, infect different cell lines and the output is robust, so it can be readily quantified (*Scaplehorn et al., 2002*; *Weisswange et al., 2009*). Taking advantage of this model we examined whether the relative positioning of pTyr motifs plays a role in the signalling output by generating a series of recombinant Vaccinia viruses expressing different A36 derivatives. We also replaced the A36 pTyr motifs with the C-terminus of the unrelated protein p14 from Orthoreovirus, which promotes N-WASP and Arp2/3 complex dependent cell fusion by interacting with Grb2 via a pTyr motif located in its short cytoplasmic tail (*Chan et al., 2020*). Interestingly, we found that as seen in A36, p14 also contains a functional Nck binding site 20 residues upstream of the Grb2 site. Moreover, the Nck binding site in both proteins plays the dominant role in promoting actin polymerisation, while Grb2 plays a supporting role. This is consistent with previous in vitro observations demonstrating that Grb2 is significantly less efficient than Nck at promoting actin polymerisation via N-WASP and Arp2/3 complex (*Carlier et al., 2000*; *Okrut et al., 2015*). Nck is essential for actin tail formation (*Scaplehorn et al., 2002*; *Weisswange et al., 2009*), however, its levels alone do not determine signal output, as its recruitment remains constant across the different A36 variants. Despite being a weak promoter of actin polymerisation, the binding position of Grb2 relative to that of Nck is an important determinant of the signalling output. Swapping the position of the two adaptor-binding sites impairs actin-based

motility and spread of the virus. The introduction of an additional Grb2 site also only leads to efficient signalling output when placed C-terminal of the Nck binding site. Grb2 appears to limit or impede the ability of Nck to stimulate N-WASP-dependent actin polymerisation.

## Optimal signalling output depends on network configuration

Nck and Grb2 are multivalent adaptors that interact with N-WASP to activate the Arp2/3 complex. Grb2 has two SH3 domains that bind N-WASP while the three SH3 domains of Nck engage with WIP and N-WASP (*Carlier et al., 2000*; *Donnelly et al., 2013*; *Rivera et al., 2004*). The linker between the first and second Nck SH3 domains also contributes to N-WASP activation (*Banjade et al., 2015*; *Bywaters and Rivera, 2021*; *Okrut et al., 2015*). These five SH3 domains and their interactions with polyproline (PxxP) motifs could, in principle, be capable of activating N-WASP via multiple routes, a notion consistent with phase transition-based signalling, which is driven by stochastic, multivalent and weak interactions (*Fuxreiter and Vendruscolo, 2021*; *Lyon et al., 2021*). Bioinformatic analysis (scan site and ELM) reveals there are 11 and 17 predicted class I and class II SH3-binding PxxP motifs in human N-WASP and WIP, respectively. We find that Grb2-binding pTyr motifs N-terminal to that of Nck lead to reduced recruitment of Grb2 and N-WASP. This suggests that there are preferred configurations of SH3-PxxP interactions to achieve optimal signalling output (actin polymerisation) via Nck/Grb2/N-WASP. Consistent with this, our previous far western analysis demonstrates that WIP and N-WASP each only contain two Nck-binding PxxP sites (*Donnelly et al., 2013*). Moreover, these sites have differing preferences for the individual SH3 domains of Nck. Recent evidence also indicates that the function of some cellular condensates such as P granules, stress granules, and puncta formed by FUS (fused in sarcoma) have underlying organisational principles (*Bienz, 2020*; *Folkmann et al., 2021*; *Jain et al., 2016*; *Kato and McKnight, 2017*; *Kato et al., 2012*). It may be, as our observations suggest, there is a degree of defined, spatial organisation within the interactions in membrane-associated signalling networks. In line with this idea, the importance of purely weak multivalent interactions in driving the assembly and function of signalling complexes has recently been extensively discussed (*Musacchio, 2022*).

## Why does signal output depend on the position of adaptor binding sites?

The Vaccinia signalling network is initiated by phosphorylation of A36 by Src/Abl family kinases (*Frischknecht et al., 1999*; *Newsome et al., 2004*; *Newsome et al., 2006*; *Reeves et al., 2005*). Unlike many systems where kinases transiently associate with their substrates, Src/Abl family kinases are constitutively associated with Vaccinia viruses undergoing actin-dependent motility. Importantly, repositioning of A36 tyrosine motifs did not impact the levels of total or activated Src kinase detected at the virus (*Figure 6A and B*). It is, however, still possible that swapping the positions of Nck and Grb2 sites leads to impairment of phosphorylation of Tyrosine 112 or 132 of A36. Multi-site phosphorylation in a protein can be processive, sometimes promoted by SH2 domains of kinases themselves (*Mayer et al., 1995*). Furthermore, kinases such as ZAP-70 and Syk have tandem SH2 domains, a regulatory feature that can be affected by tyrosine positions in their substrates (*Kulathu et al., 2019*). However, swapping Nck and Grb2-binding motifs in A36 does not detectably impair their phosphorylation (*Figure 6C and D*, *Figure 6—figure supplement 1A*). Moreover, though Grb2 is poorly recruited to the A36 G-N virus (*Figure 5A*), its binding site remains phosphorylated (*Figure 6D*). In other systems, pTyr residues that are not bound by their cognate SH2 binding partners are rapidly dephosphorylated by phosphatases (*Jadwin et al., 2018*; *Rotin et al., 1992*). Given that Grb2 has a half-life of 140 msec on the wild type virus (*Weisswange et al., 2009*), it is possible that Grb2 binds A36 G-N weakly with high turnover, thus outcompeting any dephosphorylation. Adaptor proteins like Grb2 can also promote the activity of phosphatases such as Shp2 (*Lin et al., 2021*). However, as the phosphatases involved in Vaccinia signalling are unknown, this aspect remains unexplored.

Despite being phosphorylated, repositioning of tyrosine motifs may levy spatial constraints for SH2 domain binding to A36. It is striking that the ~20 amino acid spacing between SH2-binding sites in Vaccinia A36 and Orthoreovirus p14 is also observed in the Grb2-binding Tyr171 and Tyr191 in LAT (*Huang et al., 2017*), Nck-binding Tyr1176, Tyr1193, Tyr1217 in nephrin (*Jones et al., 2006*) and Grb2-binding Tyr1092 and Tyr1110 in EGFR (*Figure 1—figure supplement 1A*). This distance could reflect structural constraints imposed by two SH2 adaptors binding nearby pTyr sites. However,

evaluation of structural data of SH2 domains bound to pTyr peptides suggests this may not be the case. The Nck1 SH2 domain has a footprint of 12 residues when bound to a phosphopeptide (EEHI-pY-DEVAADP) of Tir (PDB 2CI9), which is responsible for inducing EPEC actin pedestals (*Frese et al., 2006*). The Tir peptide has significant homology to the region surrounding Tyr112 of A36 (TEHI-pY-DSVAGSY). Furthermore, structures of Grb2-bound phosphopeptides (PDB 1TZE, PDB 1JYR) indicate that the adaptor has an even smaller footprint than that of Nck (*Rahuel et al., 1996*; *Nioche et al., 2002*). This would be consistent with space for independent SH2 binding events on A36 with no steric clash. An additional consideration is that the SH2 domain of Grb2 is unique compared to others because its cognate pTyr motif adopts a β-turn conformation when bound rather than an extended conformation (*Ettmayer et al., 1999*; *Kuriyan and Cowburn, 1997*; *Rahuel et al., 1996*). In the absence of essential structural information including that of full-length Nck, we also cannot predict whether the ability of the Grb2 SH2 domain to bind its pTyr site is influenced by the three Nck SH3 domains in the motif-swapped (A36 G-N) configuration.

It is curious that in the three viral proteins our lab has analyzed that are capable of promoting actin polymerisation via Nck and Grb2, namely A36, p14 and YL126 of Yaba-Like Disease Virus (*Dodding and Way, 2009*), Grb2 binding is C-terminal to that of Nck. It is also striking that there is no functional evidence for Grb2-binding sites upstream of Nck in any integral membrane protein. Interestingly, Grb2 is not recruited to Vaccinia in the absence of N-WASP even when its SH2-binding site on A36 is available (*Weisswange et al., 2009*). It is likely that in the A36 G-N virus, the SH3-PxxP binding between Grb2 and N-WASP is suboptimal. Taken together we favour that the configuration of SH2-binding pTyr motifs is critical for optimising downstream SH3 domain interactions that lead to actin polymerisation. This may also explain why adding extra Nck and Grb2 sites in A36 do not boost actin polymerisation (*Figure 2—figure supplement 2*, *Figure 7B*).

## Broader implications of motif positioning in networks involving disordered proteins

Our finding that specific pTyr motif configurations achieve optimal signalling output (actin polymerisation) is highly relevant to modular signalling involving disordered regions of proteins. Disordered proteins, which constitute 40% of the human proteome, are rich in short linear motifs (SLiMs) (*Tompa et al., 2014*; *van der Lee et al., 2014*; *Wright and Dyson, 2015*). Due to their ubiquity and modularity, networks assembled via SLiMs in unstructured peptides are of immense interest to biologists building synthetic signalling systems (*Lim, 2010*). In synthetic networks, the optimal organisation of globular domains is recognized to influence functionality, for example in the construction of chimeric antigen receptors (CARs) (*Finney et al., 1998*). However, the relative position of SLiMs within a complex network was not considered to play a role possibly because poorly structured polypeptides are assumed to be very flexible. More recently, CAR T-cell phenotypes were found to strongly depend on motif positioning in combinatorial libraries of non-natural SLiMs (*Daniels et al., 2022*). Future studies will confirm whether the influence of motif position on signalling output is a conserved property in other networks. Moreover, as viral proteins are enriched in disordered regions with short host-mimicking motifs (*Davey et al., 2011*; *Uversky, 2019*), they offer unique tools to explore the importance of SLiMs positioning on signalling output. Our observations also demonstrate that Vaccinia provides an excellent platform to dissect physiological or synthetic signalling networks activated by Src and Abl family kinases.

## Methods
### Expression constructs and targeting vectors
The expression vectors pE/L-LifeAct-iRFP670 (*Galloni et al., 2021*), pLVX-GFP-N-WASP (*Donnelly et al., 2013*), pE/L-GFP-Nck (SH2) (*Frischknecht et al., 1999*) and pBS SKII RFP-A3L targeting vector (*Weisswange et al., 2009*) were previously made in the Way lab. The lentiviral expression construct pLVX-GFP-Nck was generated by sub-cloning the Nck1 coding sequence (*Donnelly et al., 2013*) into NotI/EcoRI sites of a pLVX-N-term-GFP parent vector (*Abella et al., 2016*). The expression construct CB6-Src-GFP was generated by sub-cloning Src-GFP (*Newsome et al., 2006*) into BglII/NotI sites of a CB6 parent vector. All other expression constructs generated for this study were made using Gibson Assembly (New England Biolabs) according to manufacturer's instructions. Desired amino

acid substitutions were introduced by whole-plasmid mutagenesis using complementary mutagenic primers. All primers used in cloning are listed in *Table 1*. The lentiviral expression construct pLVX-GFP-Grb2 was generated by cloning a GFP-Grb2 (*Weisswange et al., 2009*) fragment into the XhoI/EcoRI sites of a pLVX parent vector (*Abella et al., 2016*). The A36R-targeting vector was generated by amplifying a fragment containing the A36R gene including 325 bp upstream and downstream sequences from the WR strain of Vaccinia virus genomic DNA and cloning into the NotI/HindIII sites of pBS SKII. This vector was modified to generate desired A36 truncations and variants. The tagGFP2 coding sequence used in the A36 N-G and A36-G-N fusion constructs was amplified from a plasmid provided by David Drubin (UC Berkeley; *Akamatsu et al., 2020*). The A36-p14 chimeric construct was obtained as a synthetic gene (Invitrogen; Geneart) and cloned into the A36R-targeting vector using SpeI/BsrGI sites in the sequences flanking the A36R coding region. A36 constructs with modified linker lengths between Nck and Grb2 binding sites were obtained as synthetic genes (IDT gBlock) and cloned into the A36R-targeting vector using KasI/BsrGI sites in the A36R coding/flanking regions. SnapGene software (from Insightful Science; available at snapgene.com) was used to plan and visualise cloning strategies, and to analyse sequencing results.

## Cell lines

HeLa cells, Nck -/- MEFs, and N-WASP -/- MEFs were provided, authenticated by STR profiling and mycoplasma-tested by the Francis Crick Institute Cell Services. All cell lines were maintained in minimal essential medium (MEM) supplemented with 10% FBS, 100 U/ml penicillin, and 100 μg/ml streptomycin at 37 °C and 5% $CO_2$. HeLa cell lines stably expressing LifeAct-iRFP670 (*Snetkov et al., 2016*) and GFP-WIP (*Weisswange et al., 2009*) were previously generated in the Way lab. Nck -/- MEFs (*Bladt et al., 2003*) and N-WASP-/-MEFs (*Snapper et al., 2001*) were provided by the late Tony Pawson (Samuel Lunenfeld Research Institute, Toronto, Canada) and Scott Snapper (Harvard Medical School, Boston, MA), respectively. For this study, lentiviral expression vectors were used to stably express GFP-Nck in HeLa cells and Nck-/- MEFs, GFP-N-WASP in HeLa cells and N-WASP-/-MEFs, and GFP-Grb2 in HeLa cells. All cell lines were generated using the lentivirus Trono group second generation packaging system (Addgene) and selected using puromycin resistance (1 μg/ml) as previously described (*Abella et al., 2016*). Expression of the relevant fusion proteins was confirmed by live imaging and immunoblot analysis (*Figure 4—figure supplement 1*, *Figure 5—figure supplement 2*). The following primary antibodies were used: anti-Nck (BD transduction; 1:1000), anti-vinculin (Sigma #V4505; 1:2000), anti-Grb2 (BD Transduction #610112, 1:3000), anti-N-WASP (Cell Signalling #4,848 S; 1:1000), anti-GFP (3E1 custom made by Cancer Research UK; 1:1000). HRP-conjugated secondary antibodies were purchased from The Jackson Laboratory.

## Viral plaque assays

Plaque assays were performed in confluent BS-C-1 cell monolayers. Cells were infected with the relevant Vaccinia virus at a multiplicity of infection (MOI)=0.1 in serum-free MEM for one hour. The inoculum was replaced with a semi-solid overlay consisting of a 1:1 mix of MEM and 2% carboxymethyl cellulose. Cells were fixed with 3% formaldehyde at 72 hr post-infection and subsequently visualised with crystal violet cell stain as previously described (*Humphries et al., 2012*). To determine plaque size, the diameter of well-separated plaques was measured using the Fiji line tool (*Schindelin et al., 2012*).

## Construction of recombinant Vaccinia viruses

In this study, recombinant Vaccinia viruses in the WR background were isolated by selecting viral plaques based on their size or by introduction of a fluorescently-tagged viral protein as described previously (*Snetkov et al., 2016*; *Weisswange et al., 2009*). The former strategy was used to generate recombinants where A36 variants were introduced at the endogenous locus by rescuing plaque size in the WR–ΔA36R virus that makes very small plaques (*Parkinson and Smith, 1994*; *Ward et al., 2003*). To introduce relevant constructs into the A36 genomic locus, HeLa cells infected with WR–ΔA36R at MOI = 0.05 were transfected with the appropriate pBS SKII A36-targeting vectors using Lipofectamine2000 (Invitrogen) as described by the manufacturer. In the case of the p14 N-G virus, a PCR fragment containing the desired construct flanked by recombination arms was used for transfection. When all cells displayed cytopathic effect at 48–72 hr post-infection, they were lysed,

**Table 1.** Primers.

| No. | Sequence | Construct(s) generated |
| --- | --- | --- |
| AB004 | CCAGCAACACTATCGTAAATGTTCTGTATTACGATCATTATTTATTAGCAG | A36 N-N |
| AB007 | ACACATTTTCGATAGTGTTGCTGG | A36 X-G |
| AB008 | GCAACACTATCGAAAATGTGTTCTG | A36 X-G |
| AB009 | CAGACTATTTTTCAGAACACTACAGTAGTA | A36 N-X, A36 X-N, A36 X-X |
| AB010 | CTGTAGTGTTCTGAAAAATAGTCTGT | A36 N-X, A36 X-N, A36 X-X |
| AB105 | TCAGCCAGCACGAGTTCGAGGACCCCTACGAGCCCCCAG | p14 N-G (Y96F) |
| AB106 | CTGGGGGGCTCGTAGGGGTCCTCGAACTCGTGCTGGCTGA | p14 N-G (Y96F) |
| AB107 | ACGAGTACGAGGACCCCTTCGAGCCCCCCAGCAGGAGGAA | p14 N-G (Y100F) |
| AB108 | TTCCTCCTGCTGGGGGGCTCGAAGGGGTCCTCGTACTCGT | p14 N-G (Y100F) |
| AB109 | CTACAGCACCTTCGTGAACATCGACAACGTGAGCGCCATC | p14 N-G (Y116F) and p14 N-G (Y96,116F) |
| AB110 | GCGCTCACGTTGTCGATGTTCACGAAGGGTGCTGTAGGGG | p14 N-G (Y116F) and p14 N-G (Y96,116F) |
| AB040 | TACCGGACTCAGATCTCGAGATGAGCAAGGGCGAGGAGC | pLVX-GFP-Grb2 |
| AB057 | CGTCGACTGCAGAATTCTTACTAGACGTTCCGGTTCACGGGGGG | pLVX-GFP-Grb2 |
| AB051 | CACCGCGGTGGGCGGCCGCCGCTCATCATAGCAT | A36 N-G, A36 G-N, A36 G-N-G, A36 G-G-N |
| AB052 | GGTCGACGGTATCGATAAGCTTTATCTATAGAGATAACAC | A36 N-G, A36 G-N, A36 G-N-G, A36 G-G-N, p14-N-G, p14 G-N, A36 G-G, A36 N-N |
| AB053 | TGATTAGTTTCCTTTTTATAAAATTGAAGTAATATTTAGT | A36 N-G, A36 G-N, A36 G-G |
| AB054 | ATAAAAAGGAAACTAATCACGTGCTTCCAGCAACACTAT | A36 G-N |
| AB055 | ATAAAAAGGAAACTAATCAAATTACTACTGTAGTGTTCTG | A36 N-G |
| AB087 | TATTTATCAGAACACTACAGTAGTAATTTGATTAGTTTCCTTTTTA | A36 G-N-G |
| AB088 | AGTGTTCTGATAAATAGTCTGTTCATTACGATCATTATTTATTAGCAGCGGTGCTTCCAGCAACA | A36 G-N-G |
| AB089 | CTAATAAATAATGATCGTAATGAACAGACTATTTATCAGAAC | A36 G-G-N |
| AB090 | ATTACGATCATTATTTATTAGCAGAATTACTACTGTAGTGTTCTGATAAAT | A36 G-G-N |
| AB096 | AAACAATAAAATATTGAACTAGTAGTACGTATATTGAGC | A36 N-G-TagGFP2, A36 G-N-TagGFP2, p14 N-G, p14 G-N, A36 N-N, A36 G-G |
| AB097 | TCCGGTGGCGACCGGTGGATCCGACCCCAATTACTACTGTAGTGTTCTGATAA | A36 N-G-TagGFP2 |
| AB098 | TCCGGTGGCGACCGGTGGATCCGACCCCGTGCTTCCAGCAACACTATCGTAA | A36 G-N-TagGFP2 |
| AB099 | ACAGAACACATTTACGATAGTGTTGCTGGAAGCACGTGATTAGTTTCCTTTTTATAA | A36 N-N |
| AB104 | TATAAAAAGGAAACTAATCAAATTACTACTGTAGTGTTCTGATAAAATAGTCTGTTCATTACGATC | A36 G-G |
| AB121 | AGGAACAGCTACAGGCTGAG | p14 N-G, p14 G-N |
| AB122 | CTCAGCCTGTAGCTGTTCCTCGCCATGACATTGGATT | p14 N-G, p14 G-N |

and serial dilutions of the lysates were used to infect confluent BS-C-1 cell monolayers in a plaque assay (see above). Plaques were revealed by neutral red staining and recombinants were identified and picked based on increased plaque size. Plaque lysates were used to infect fresh BS-C-1 cell monolayers over at least three rounds of plaque purification. To isolate A36 N-G-TagGFP2 and A36 G-N-TagGFP2 viruses, in addition to size, plaque fluorescence was used to identify recombinants. To generate recombinants where the viral core was fluorescently labelled with RFP, HeLa cells infected with the relevant parent virus were transfected with the pBS SKII RFP-A3-targeting vector (*Weiss-wange et al., 2009*). Recombinant viruses were isolated based on RFP fluorescence over at least three rounds of plaque purification. In all cases, successful recombination at the correct locus, loss of the parent variant and virus purity were verified by PCR and sequencing. Plaque sizes of viruses obtained from independent recombination events were also compared where possible, to control for effects arising from off-target mutations (*Figure 2—figure supplement 1C*, *Figure 7—figure supplement 2*). For *Figure 2—figure supplement 2B* and *Figure 3—figure supplement 1A*, experiments were performed using crudes lysates from cells infected with indicated recombinants. For all remaining experiments, recombinant viruses were purified through a sucrose cushion before use and storage.

## Vaccinia virus infection for imaging

For live and fixed cell imaging, cells plated on fibronectin-coated MatTek dishes (MatTek corporation) or coverslips were infected with the relevant Vaccinia virus recombinant in serum-free MEM at MOI = 1. After one hour at 37 °C, the serum-free MEM was removed and replaced with complete MEM. Cells were incubated at 37 °C until further processing.

## Transient transfection and siRNA

Transient transfection of A36-p14 hybrid constructs, CB6-Src-GFP, pE/L-GFP-Nck (SH2) and the pE/L--LifeAct-iRFP670 expression in Vaccinia-infected cells was done using FUGENE (Promega) as described by the manufacturer. To transiently express the A36-p14 chimera and its variants (*Figure 3B-E*), expression vectors containing the relevant construct under the control of the A36 promoter were transfected into cells one hour after infection with the WR-ΔA36 virus (*Parkinson and Smith, 1994*). pE/L-LifeAct-iRFP670 (*Figures 4A and 6A*) and pE/L-GFP-Nck(SH2) (*Figure 6B*) were transfected into cells one hour after infection with the relevant viruses. CB6-Src-GFP (*Figure 6A*) was transfected into cells 16 hr prior to infection. For knockdown experiments, HeLa cells were transfected with siRNA as previously described (*Abella et al., 2016*). Cells were infected with Vaccinia virus 72 hr after siRNA transfection, and samples from each siRNA condition were kept for immunoblot analysis. The following siRNAs were used: AllStars (Qiagen; SI03650318), Grb2-targeting siRNA oligos AGGC CGAGCGUAAUGGUAA, GAAAGGAGCUUGCCACGGGUU and CGAAGAAUGUGAUCAGAACUU. The following primary antibodies were used in immunoblots: anti-vinculin (Sigma #V4505; 1:2000), anti-Grb2 (BD Transduction #610112, 1:3000). HRP-conjugated secondary antibodies were purchased from The Jackson Laboratory.

## Immunofluorescence

At 8 hr (HeLa) or 16 hr (MEFs) post-infection, cells were fixed with 4% paraformaldehyde in PBS for 10 min, blocked in cytoskeletal buffer (1 mM MES, 15 mM NaCl, 0.5 mM EGTA, 0.5 mM MgCl$_2$, and 0.5 mM glucose, pH 6.1) containing 2% (vol/vol) fetal calf serum and 1% (wt/vol) BSA for 30 min, and then permeabilised with 0.1% Triton-X/PBS for 5 min. To visualise cell-associated enveloped virions (CEV), cells were stained with, with a monoclonal antibody against B5 19C2, rat, 1:1000; (*Schmelz et al., 1994*) followed by an Alexa Fluor 647 anti-rat secondary antibody (Invitrogen; 1:1000 in blocking buffer) prior to permeabilisation of the cells with detergent. Other primary antibodies used were anti-Nck (Millipore #06–288; 1:100), anti-WIP 1:100; (*Moreau et al., 2000*), and anti-N-WASP (Cell Signalling #4,848 S; 1:100) followed by Alexa Fluor 488 conjugated secondary antibodies (Invitrogen; 1:1000 in blocking buffer). Actin tails were labeled with Alexa Fluor 488, Alexa Fluor 568 or Alexa Fluor 647 phalloidin (Invitrogen; 1:500). Coverslips were mounted on glass slides using Mowiol (Sigma). Coverslips were imaged on a Zeiss Axioplan2 microscope equipped with a 63 x/1.4 NA Plan-Achromat objective and a Photometrics Cool Snap HQ cooled charge-coupled device camera. The microscope was controlled with MetaMorph 7.8.13.0 software. To measure the levels of A36 pY132 and Src pY418 at the virus (*Figure 6*) coverslips were fixed with 4% paraformaldehyde containing

0.1% Triton-X prior to staining with an antibody against the A36 phosphotyrosine 132 site 1:100, (*Newsome et al., 2004*) or the Src phosphotyrosine 418 site (1:300, Life Technologies #44,660 G). Mounted coverslips were imaged on an Olympus iX83 Microscope with Olympus 100 x/1.50NA A-Line Apochromatic Objective Lens, dual Photometrics BSI-Express sCMOS cameras and CoolLED pE-300 Light Source. The microscope was controlled with MicroManager 2.0.0 software.

### Live-cell imaging

Live-cell imaging experiments were performed at 8 hr (HeLa) or 16 hr (MEFs) post-infection in complete MEM (10% FBS) in a temperature-controlled chamber at 37 °C. Cells were imaged on a Zeiss Axio Observer spinning-disk microscope equipped with a Plan Achromat 63 x/1.4 Ph3 M27 oil lens, an Evolve 512 camera, and a Yokagawa CSUX spinning disk (*Galloni et al., 2021*; *Pfanzelter et al., 2018*). The microscope was controlled by the SlideBook software (3i Intelligent Imaging Innovations). For determining recruitment levels of GFP-tagged molecules to the virus, single snapshots of live cells were acquired. To determine virus speed, images were acquired for 50 s at 1 Hz.

### Image analysis and quantitation

Quantification of actin tail number and length was performed using two-colour fixed cell images where actin and extracellular virus were labelled. Ten cells were analysed per condition in each independent experiment. The number of actin tails was measured by blindly selecting 25 isolated extracellular virus particles in each image and determining the presence of a tail in the corresponding actin channel. Actin tail length of 8 randomly selected tails per image was measured using the freehand line drawing function in Fiji.

To analyse virus motility, two-colour time-lapse movies of HeLa cells stably expressing LifeAct-iRFP670 infected with the relevant recombinant virus labelled with RFP-A3 were used. The velocity of virus particles in the RFP channel was measured using a Fiji plugin developed by David Barry (the Francis Crick Institute) as previously described (*Abella et al., 2016*). Bona fide actin-based virus motility was verified manually using the corresponding iRFP670 channel. Five movies were analysed per condition, and speeds from 30 particles were measured in each independent experiment.

A36 pY132 and Src pY418 antibody intensities were analysed in two-colour fixed cell images where actin was co-labelled. Raw integrated density of the antibody signal was measured at the tip of actin tails after local area-corrected background subtraction in Fiji as detailed elsewhere (*Verdaasdonk et al., 2014*). Thirty particles were measured per condition in each independent experiment.

Recruitment levels of GFP-tagged molecules to the virus were analysed in two-colour live cell images as an intensity ratio to RFP-A3. Five images were analysed per condition, and 25–30 particles were measured in each independent experiment. Only cells with comparable expression levels of GFP-tagged proteins were used; overexpressing cells were excluded. To focus on late infection stages of viral egress, cells with many particles at the periphery were selected. Particles with RFP-A3 intensity of greater than 3,500 raw grayscale units were excluded because under our imaging conditions these values typically corresponded to two or more overlapping virus particles. GFP images were background subtracted using a median filtered image. The ratio of GFP:RFP intensity was then measured using Fiji at isolated peripheral particles that discernably showed actin-based motility when visualised live. For quantification of GFP-Nck (SH2) in Nck null cells (which cannot form actin tails), particles with any GFP recruitment were used for measurement. In the recombinant viruses we analysed, the percentage of virus particles forming actin tails and recruiting signalling components did not change. Furthermore, by measuring GFP-Nck intensity at extracellular virus particles in fixed cells, it was independently verified that the measurements made as a ratio to RFP A3 intensity as described above, reflected results seen on bona fide CEVs.

### Phosphopeptide pulldown assay

Phosphorylated and non-phosphorylated peptides were synthesised in-house (p14 Y96: SQHEpYED-PYEPP, SQHEYEDPYEPP; A36 Y112: APSTEHIpYDSVAGST, APSTEHIYDSVAGST; p14 Y100: YEDPpY-EPPSRRK, YEDPYEPPSRRK) containing the predicted Nck-binding sites and an N-terminal biotin tag. These were coupled to streptavidin Dynabeads M-280 (Thermo Fischer Scientific). Uninfected HeLa cells were lysed in a buffer containing 50 mM Tris.HCl pH7.5, 150 mM NaCl, 0.5 mM EDTA, 0.5% NP40, 0.5% Triton-X and a cocktail of protease and phosphatase inhibitors (1 mM orthovanadate, cOmplete

(Roche), PHOSstop (Roche)). A postnuclear supernatant was obtained by a 16,000 g centrifugation for 10 min at 4 °C. The peptide-coupled beads were incubated with these clarified cell lysates. Unbound proteins were removed from beads in three washes in the same cell lysis buffer. The proteins bound to beads were resolved on an SDS-PAGE and the presence of Nck was determined by immunoblot analysis using anti-Nck antibody (Millipore #06–288; 1:1000). As a negative control anti-Grb2 (BD Transduction #610112; 1:3000) was used. HRP-conjugated secondary antibodies were purchased from The Jackson Laboratory.

### Whole-cell lysate phosphoblot

HeLa cells infected with A36 N-G-TagGFP2 or A36 G-N-TagGFP2 were lysed at 9 hr post-infection in PBS containing 1% SDS, a cocktail of protease and phosphatase inhibitors (1 mM orthovanadate, cOmplete (Roche), PHOSstop (Roche)) and Benzonase (Millipore). Proteins from these lysates were resolved on an SDS-PAGE and the presence of total pTyr and A36 were determined by immunoblot analysis using anti-phosphotyrosine PY99 antibody (Santa Cruz #sc-7020; 1:1000) and anti-TagGFP2 antibody (Evrogen #AB121; 1:3000) respectively. As a loading control anti-Grb2 (BD Transduction #610112; 1:3000) was used. HRP-conjugated secondary antibodies were purchased from The Jackson Laboratory.

### Statistical analysis and figure preparation

All data are presented as means ±S.D. For all experiments, means of at least three independent experiments (i.e. biological replicates) were used to determine statistical significance by a Welch's t-test (comparing only two conditions), Tukey's multiple comparisons test (comparing multiple conditions with each other) or a Dunnett's multiple comparisons test (comparing multiple conditions with a control). All data are represented as SuperPlots to allow assessment of the data distribution in individual experiments (*Lord et al., 2020*). SuperPlots were generated using the SuperPlotsOfData webapp (*Goedhart, 2021*) and graphs showing intrinsic disorder predictions were generated in GraphPad Prism 9. All data were analyzed using GraphPad Prism 9 or the SuperPlotsOfData webapp. Temporal overlays of live imaging data to illustrate virus motility were generated using the temporal colour-code function in Fiji. Schematics were created with BioRender.com. Final figures were assembled using Keynote software.

## Acknowledgements

We thank Nicola O'Reilly, Dhira Joshi and Stefania Federico (Peptide Chemistry, the Francis Crick Institute) for synthesizing peptides. We thank Cell Services and Genomics Equipment Park at the Francis Crick Institute for their help with maintaining cell lines and DNA sequencing respectively. We thank members of the Way Laboratory for useful discussions and suggestions, in particular Davide Carra, for nucleating the idea of adding an extra Grb2 site, and Alessio Yang for help in generating the Src-GFP construct. We also thank Frank Uhlmann and Neil McDonald (the Francis Crick Institute) for helpful comments on the manuscript. Michael Way was supported by Cancer Research UK (FC001209), UK Medical Research Council (FC001209), and Wellcome Trust (FC001209) funding at the Francis Crick Institute. For the purpose of Open Access, the authors have applied a CC BY public copyright license to any Author Accepted Manuscript version arising from this submission.

## Additional information

### Funding

| Funder | Grant reference number | Author |
| --- | --- | --- |
| Cancer Research UK | FC001209 | Michael Way |
| Medical Research Council | FC001209 | Michael Way |
| Wellcome Trust | FC001209 | Michael Way |

| Funder | Grant reference number | Author |
|--------|------------------------|--------|

The funders had no role in study design, data collection and interpretation, or the decision to submit the work for publication. For the purpose of Open Access, the authors have applied a CC BY public copyright license to any Author Accepted Manuscript version arising from this submission.

## Author contributions

Angika Basant, Conceptualization, Formal analysis, Investigation, Visualization, Methodology, Writing - original draft, Writing - review and editing; Michael Way, Conceptualization, Supervision, Funding acquisition, Project administration, Writing - review and editing

## Author ORCIDs

Angika Basant http://orcid.org/0000-0002-4754-6647
Michael Way http://orcid.org/0000-0001-7207-2722

## Decision letter and Author response

Decision letter https://doi.org/10.7554/eLife.74655.sa1
Author response https://doi.org/10.7554/eLife.74655.sa2

# Additional files

## Supplementary files

• Transparent reporting form

## Data availability

All data generated or analysed during this study are included in the manuscript and supporting file; Source Data files have been provided for all graphs and western blots.

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
