## [Editor Report]

The authors have previously established that the binding of the NCK and GRB2 SH2/SH3 adaptor proteins to their cognate pTyr sites in the C-terminal cytoplasmic domain of the viral A36 protein embedded in the Vaccinia virion membrane is important for the formation of actin tails on the virion that drive intracellular virus motility and cell to cell spread. Here, they made the surprising observation that it is essential to have the NCK-binding site upstream of the GRB2-binding site for the formation of functional actin tails. This suggests that precise spatial organization of signaling protein complexes that drive actin cytoskeleton assembly is key to optimal signal output, and, by extension, this principle may be important in other signaling pathways with multiple inputs.

---

## [Decision Letter]

**Decision letter after peer review:**

Thank you for sending your article entitled "The Relative Binding Position of Nck and Grb2 Adaptors Dramatically Impacts Actin-Based Motility of Vaccinia Virus" for peer review at *eLife*. Your article is being evaluated by 4 peer reviewers, including Tony Hunter as Reviewing Editor and Reviewer #1, and the evaluation is being overseen by Anna Akhmanova as the Senior Editor.

All the reviewers agree that your finding that the NCK and GRB2 pTyr-binding sites in the IDR of the Vaccinia A36 protein have to be in a defined order in order to promote effective actin tail formation and virus motility is an interesting and to some extent unexpected observation. However, as you will see from the reviews, the consensus is that at least some additional mechanistic insights into this observation are needed before your paper would be suitable for publication in *eLife*. The reviews include a number of experimental suggestions for how some clues to mechanism might be obtained. However, before requesting a revised version, we would like you to develop and submit a plan for additional experiments that you could do within a reasonable time frame to provide some further insights into mechanism underlying the required order of binding sites. On the basis of your response a decision will be made whether to request a revised version. We recognize that some of the experiments are likely to require generation of new recombinant viruses, which will be time consuming, and this will be taken into account in making the decision. We hope it will be possible for you to provide further mechanistic insights into your intriguing observations so that we can consider your paper further for *eLife*.*Reviewer #1:*

Here the authors have investigated whether the relative positions of the pTyr binding sites for the NCK and GRB2 SH2 domains that bind to Tyr112 and Tyr132, respectively, present in a disordered region in the A36 Vaccinia virus integral viral membrane protein. These sites are phosphorylated by the cellular SRC and ABL tyrosine kinases during Vaccinia virus infection cycle in target cells, and are important for triggering local polymerization of the actin cytoskeleton and actin-dependent cell-to-cell spread of virus. For this purpose, they used a system in which the WR-ΔA36R Vaccinia virus strain lacking the A36 gene was rescued by expression of the A36 N-G variant that terminates after the GRB2 binding site at aa 139, but can still activate actin polymerization, in a manner dependent on the Y112 NCK bindng site, and also mediate microtubule transport via the upstream kinesin-1 bindng sites. They started by swapping the 12 aa sequences sround the NCK pY112 and GRB2 Y120 binding sites generating the A36 G-N variant, and showed that A36 G-N induced significantly shorter F-actin tails than A36 N-G. Next, they made A36 N-X and X-N variants, which both lack a GRB2 binding site, and found these induced F-actin tails of the same length as the corresponding A36 N-G and A36 G-N variants, indicating that the Y112 position of the NCK binding site in A36 is important and sufficient for F-actin tail formation. Consistently, the A36 G-N variant showed reduced virus motility and cell-to-cell spread compared to the WT A36.

To determine the generality of these findings, they devised a synthetic pTyr network in which the N-G region of A36 is replaced with the corresponding region of a different viral protein capable of activating actin polymerization via the NCK and GRB2 adaptors, namely p14, an orthoreovirus integral membrane protein that activates N-WASP via GRB2 bound to pY116 to regulate cell fusion. P14 has two additional Tyr residues 16 and 20 aa upstream of Y116 that have contexts matching NCK SH2 domain binding sites. In the chimera, aa 1-105 of A36 are fused to aa 79-125 of p14, and they showed that A36-p14 N-G induced the formation of F-actin tails efficiently, and that this required N-WASP. Mutation of Y96, a putative NCK binding site, but not Y100 in A36-p14 N-G reduced F-actin tail formation, and the combined Y96/Y116 mutation reduced this further. When HeLa cells were infected with a recombinant Vaccinia virus with A36 replaced by A36-p14 N-G they showed that NCK, WIP and N-WASP as well as GFP-GRB2 were recruited to viriions and induced F-actin tail formation, whereas an A36-p14 G-N virus failed to induce F-actin tails or cell-to-cell spread. The authors concluded that the N-G organization of the two key Tyr phosphorylation sites that recruit NCK and GRB2 is a general feature of viral proteins that initiate F-actin formation to drive virion movement.

Finally, the authors carried out further experiments to elucidate why the N-G site organization is important for inducing proper F-actin reorganization. First, they showed that the A36 N-G and A36 G-N proteins were expressed at the same level in infected cells. Using GFP-tagged NCK, GRB2, WIP and N-WASP, they also found that the levels of recruited NCK were comparable between the A36 N-G and G-N viruses in infected HeLa cells, whereas the levels of GRB2 and N-WASP recruitment were significantly reduced on the A36 G-N virus. Using an A36 N-X virus for infection, they showed that reduction in GRB2 binding did not affect the level of NCK binding, although there was some decrease in N-WASP binding, but less than that observed with A36 G-N. Second, by staining with an anti-pY132 antibody that detects the phosphorylated A36 GRB2 bindng site, they showed that level of pY132 was significantly higher in the A36 G-N than the A36 N-G virus, whereas the anti-pY132 antibodies did not detect the A36 N-X virus. They also found that siRNA knockdown of GRB2 increased pY132 labeling on the A36 N-G virus. Finally, knowing that the position of the GRB2-binding site relative to the NCK binding site is critical for the output of the A36 signaling network, the authors tested whether an additional GRB2 binding site would influence the output by generating recombinant A36 G-N viruses with an extra GRB2 site either N-terminal (A36 G-G-N virus) or C-terminal to the NCK site (A36 G-N-G virus). All four viruses (N-G, G-N, G-N-G and G-G-N) recruited similar levels of NCK and induced comparable numbers of actin tails. However, while the N-G and G-N-G viruses recruited similar levels of N-WASP, these were ~2-fold greater than for the G-N and G-N-G viruses; the actin tail lengths formed by the A36 G-N and G-G-N viruses were equally short, with the G-N-G virus were noticeably longer, but not as long as the N-G virus. The N-G and, particularly, the G-N-G viruses also recruited significantly more GFP-GRB2 than the G-N and G-G-N viruses. They concluded that GRB2 binding to A36 is not essential for Vaccinia infection to induce actin tail formation, but that its binding position on A36 relative to that of NCK influences the level of N-WASP recruitment to promote proper F-actin reorganization.

The finding that the N- to C-terminal order of the NCK and GRB2 pTyr binding sites in the cytoplasmic tail of the A36 Vaccinia virus membrane protein is important for proper activation of local actin polymerization and formation of actin tails, and subsequent virion movement, is interesting, particularly in light of the recent realization that signaling through phosphorylated IDRs is often mediated by formation of local biomolecular condensates that create signaling nodes. However, based on their results, the authors argue that A36-mediated signaling is unlikely to be a simple phase transition mechanism triggered by increased local concentrations of the NCK and GRB2 signaling components, as has been proposed for other signaling biocondensates, and that instead the A36 network has underlying wiring principles with a preferred configuration or directionality. While the data provide convincing evidence that there is a required order of A36 pTyr binding sites,, in the end the results do not provide true mechanistic insights into why this particular binding site order is important.

Overall, these findings will be of interest to people working on signaling biocondensates formed on phosphorylated IDRs, and those studying mechanisms of Vaccinia virus infection and cell to cell movement. Some simple experiments could provide further mechanistic insights, such as varying the distance between the NCK and GRB2 binding sites, determining phosphorylation stoichiometry of the two sites and whether both sites are simultaneously phosphorylated in a single A36 molecules. Further insights into the logic of the network might be gained by biochemical and possibly structural efforts to reconstitute the system using different forms of the phospho-A36 C-terminal region with purified WT and mutant NCK, GRB2, WIP and N-WASP proteins, although admittedly such experiments might be beyond the scope of this paper.

Recommendations for the authors:

Based on the model in Figure 1, it is implicit in the authors' thinking that both NCK and GRB2 can bind simultaneously to a single diphosphorylated pY112/pY132 A36 molecule, but they provide no evidence that this is actually the case, or even evidence that both Y112 and Y132 sites are phosphorylated in a single A36 molecule (digestion of phospho-A36 with a Lys-specific protease, which will put both Y112 and Y132 in the same peptide, followed by MS analysis, perhaps with NCK or GRB2 SH2 domain pre-enrichment, might provide such evidence). If NCK and GRB2 can indeed bind simultaneously to the same A36 molecule, this raises the question of whether the two bound proteins physically interact, and whether such an interaction might have functional consequences. Extending this idea, why then would switching the order of the two pTyr sites reduce signal output? Here, one should note that the binding of the SH2 domains of NCK and GRB2 to their target pTyr sites is directionally oriented, and by reversing the order of the sites such protein-protein interactions might not be able to take place properly. In this regard, perhaps surprisingly, the authors did not test whether the spacing between the two pTyr sites is critical (i.e. does it have to be 20 amino acids? N.B. The spacing between the two pTyr residues in ITAMs is 11 aa, and these bind tandem SH2 domains), which could be the case if specific interactions between the bound NCK and GRB2 proteins are required. Here, it is notable that the spacing of the functional NCK and GRB2 binding sites in the orthoreovirus p14 protein is also 20 amino acids. Although the authors showed that loss of GRB2 binding did not reduce NCK binding, which implies that their binding is not cooperative, it would certainly be informative to test the effects of increasing or decreasing the spacing of the NCK and GRB2 SH2 binding sites on the output of the A36 network.

It might also be worth carrying out structural modeling of a diphosphorylated WT A36 105-140 aa peptide bound simultaneously to an NCK SH2 domain (plus the neighboring SH3 domain?) and a GRB2 SH2 domain (plus the neighboring SH3 domains?) to see if any structural constraints emerge, and whether any differences are apparent when the phosphorylated G-N version of this peptide is used. Such analysis might provide further insights into the preferred directionality.

Points: 1. Figure 2: Did the authors show that the GRB2 Y112 site and NCK Y122 sites in the A36 G-N protein are phosphorylated to the same extent as in A36 N-G protein? In Figure 6, they used anti-pY132 antibodies to demonstrate that phosphorylation of the pY132 GRB2 site was similar or even higher in the G-N protein than the parental N-G protein, but apparently did not do this for the pY112, perhaps due to the lack of suitable phospho-specific antibodies. Instead, it might be possible to use GST-NCK SH2 domain and GST-GRB2 SH2 domain overlay blots of A36 N-G and G-N proteins isolated from infected cells for this purpose.

2. Figure 3F: The authors used phosphopeptide pulldowns to show that a p14 pY96 peptide pulled down NCK and that a p14 pY116 peptide pulled down GRB2, but it appears that they did not test if a p14 pY100 peptide could pull down NCK, which seems important. Even though the Y100F mutant form of A36-p15 did not exhibit a deleterious phenotype, it is unclear whether Y100 is in fact phosphorylated in infected cells, and, more importantly, whether it can recruit NCK if it is phosphorylated, e.g. in a Y96F/Y116F mutant background. As indicated above, it is possible that the shorter 16-residue spacing between pY100 and pY116 is incompatible with appropriate signal output.

3. Figure 4: Did the authors demonstrate directly that both Y96 and Y116 in the A36-p14 N-G fusion are phosphorylated and that A36-p14 N-G binds both NCK and GRB2?

4. Figure 6: The fact that siRNA knockdown of GRB2 increased pY132 antibody labeling on the A36 N-G virus is interesting. However, the authors should note that SH2 domain binding is reported to increase pTyr levels of target residues in vivo (e.g. see PMID: 1537335), presumably because the SH2-bound pTyr residues are protected from dephosphorylation by PTPs. If this is also the case for GRB2, then one might have expected the siRNA-mediated reduction in GRB2 levels to cause a decrease in pY132 levels, rather than an increase as observed. This issue would be worth some discussion.

5. Figure 7: The authors should indicate in the text that spacing of the extra GRB2 sites in A36-G-N-G and G-G-N viruses was also 20 aa.

6. Did the authors make and test an A36 N-N virus? This might generate too "strong" a signal and perturb the output. A pertinent example here would be ES cells in which the key FGFR-GRB2/SOS-RAS-ERK pathway is disrupted when endogenous GRB2 is replaced by a GRB2 with a superbinder SH2 domain that causes pathway hyperactivation and thereby prevents primitive endodermal lineage formation (PMID: 23452850).

7. A propos whether both Y112 and Y132 are simultaneously phosphorylated in a single A36 molecule, is the stoichiometry of phosphorylation at these two sites known. Also, would it be possible for two neighboring A36 molecules in the outer membrane of a single virion to collaborate if Y112 is phosphorylated in one A36 molecule and Y132 phosphorylated in the second A36 molecule?*Reviewer #2:*

The authors use the Vaccinia virus system to examine the potential role of the position of Nck and Grb2 binding sites within the unstructured viral A36 protein in the assembly of a functional actin comet tail. An advantage of these studies is the Way group's extensive expertise in this system, one of the first and perhaps still one of the best systems for studying how extracellular signals can lead to the assembly of dynamic actin structures through localized activation of N-WASP. They have previously characterized the specific roles of Nck (the primary driver of actin assembly) and Grb2 (which plays a supporting role) in the number, size, and speed of resulting actin structures. Here the goal is to test whether the positioning of the phosphorylated Nck and Grb2 SH2 binding sites affects any of these parameters. They are able to show convincingly that positioning the Nck binding site in an N-terminal (membrane proximal) position leads to more robust actin tail formation (measured primarily by length of the tail, its speed, and plaque size of the recombinant vaccine virus), compared to constructs where the Grb2 binding site is N-terminal.

This result is quite surprising, given the expectation that the A36 intracellular region in unstructured, and raises interesting questions as to the mechanism responsible. They provide fairly strong evidence that viruses with compromised actin assembly also have reduced Grb2 (as well as N-WASP) binding, despite robust phosphorylation of the Grb2 binding site, suggesting that this is primarily responsible for the decreased signal output. They also speculate that this system provides novel insight into the still mysterious design principles underlying phase-separated molecular condensates on the membrane, an area of intense current interest in the fields of cell biology and cell signaling.

In general the experiments are well designed and rigorously interpreted. However the somewhat modest effects, in my view, lessen the impact of the results. As this group showed many years ago, binding of Nck to A36 is the major driver of actin assembly in this system, while Grb2 binding has modest effects on the length and speed of the resulting tails (less than 2-fold difference). Thus the dynamic range of experiments where the positioning of Nck and Grb2 binding sites is manipulated is relatively small. One could legitimately ask if a less than two-fold difference in actin tail length is that biologically meaningful, and whether this is a sufficient basis for extrapolating to the general behavior of membrane-associated molecular condensates.

A larger issue in my mind is that even if one accepts that the differences in output based on positioning of adaptor binding sites is biologically significant, the results don't provide a mechanistic model or even a testable hypothesis for the effect. Thus the impact on our understanding of cell signaling mechanisms in general, or more specifically systems where clustered receptors and their binding proteins may have properties of phase-separated molecular condensates, is rather limited. This is a very interesting and intriguing observation to be sure, but without a mechanistic basis I don't think it advances the field significantly.

Recommendations for the authors:

In my view the most serious concerns with the data as presented relate to mechanisms that might explain the apparent decrease in Grb2 binding when it is positioned N-terminal to the Nck binding site. I think the simplest explanation is that for some reason Grb2 binding sites are not as well phosphorylated when positioned N-terminally. This could be due to decreased accessibility to the kinases, increased accessibility to phosphatases, decreased accessibility of the phosphorylated site to Grb2, or many other possibilities. The experiments that show robust phosphorylation of the N-terminal Grb2 site (Figure 6) are compromised by several factors. First, only a phosphospecific antibody to the Grb2 site is used in these experiments. Similar experiments using a phosphospecific Nck binding site antibody in parallel would control for a number of issues, including whether position consistently affects relative steady state phosphorylation of both sites.

Furthermore, while I understand that the authors are most interested in the phosphorylation state of A36 directly under virus particles (where actin tails are assembling), it would be very informative to see results from simple immunoblots of whole cell lysates to assess bulk phosphorylation of Nck and Grb2 sites in cells infected with various viruses. The argument that the apparent increase in Grb2 binding site phosphorylation might be due to decreased shielding from antibody of sites by bound Grb2 (p. 10) is not very compelling to me, especially since Grb2 knockdown has a very modest (and statistically insignificant) effect on antibody binding to this site. Furthermore, the authors do not account for the effect of bound SH2 domains in increasing overall phosphorylation of their binding sites by protecting them from dephosphorylation; this would tend to oppose any apparent decrease in antibody accessibility due to SH2 binding.

In general, I think it would have been helpful to compare results for various constructs with those for the N-X virus (completely lacking the Grb2 site), in order to assess their activity relative to when Nck binding is maximal and Grb2 binding is abolished. However I don't believe this is essential, given the amount of work required to re-do experiments.*Reviewer #3:*

The authors investigated the modularity of the viral protein A36, which mediates actin-based motility of vaccinia virus. Tyrosine phosphorylation of two sites on A36 recruit the adaptor proteins Nck and Grb2, ultimately leading to activation of the actin nucleator N-WASP. They manipulated the order of the Nck and Grb2 binding sites in recombinant viruses carrying a minimal A36 backbone and examined the efficiency and speed of actin nucleation in infected cells. They find that the order of these two sites within a largely unstructured region of the protein plays an important role in governing signal strength, and swapping these motifs attenuated virus motility. They demonstrated that these findings might be broadly applicable to how cell signalling circuits are wired by replacing the Nck and Grb2 sites with sites from an unrelated reovirus protein that signals in a mechanistically similar fashion, and found identical results.

The authors are able to to build on a very well described signalling module that enables cell signalling to be dissected in vivo using powerful pathogen and host genetics. All the manipulations to binding sites were performed with recombinant viruses allowing correlations of virus motility, signalling, adaptor recruitment and virus spread. There is robust quantification of the data provided and high quality imaging. Extending their analysis to the reovirus p14 actin nucleator demonstrates the applicability of this research to the field of synthetic biology, and the plastic nature of these circuits.

A limitation of the study is that every manipulation to A36 effectively compromises the function of the protein. The ability to predict modifications that enhance signalling output will reveal true mastery of this signalling cassette.

Recommendations for the authors:

A key question is whether the authors are assessing exclusively the impact of the order of sites, or whether some sites are more or less efficient within the context of the full-length protein. Figure 2B Suppl 1 may indicate the authors' interpretation is not as straightforward as it seems, all three replicates reveal a downward trend in actin tail length when comparing A36-NX with A36-X-N. While not significant according to the authors' statistical test, this may be a question of sensitivity. This suggests that the difference between the two sites is not as black and white as the authors interpret. Conversely, when it suits the authors' hypotheses, they choose to interpret non-significant differences as supporting their claims (increased pY132 labelling in 6B). I recommend couching the interpretation of Figure 2B Suppl 1 to include the possibility that the sites may not be equivalent or provide data that better supports the authors' interpretation.

Some details on the validation of recombinant viruses appears to be missing. Were multiple independent clones testing to exclude the possibility that selected viruses didn't carry extraneous (off-target) mutations. The information that multiple clones had similar plaque sizes, for example, would strengthen the analysis.

In general, the manuscript was clear, concise and well written. I do think the sentence on L274 should be reworded: "This increase was less dramatic than the G-N virus as the knockdown was not complete and the virus is good at recruiting any remaining Grb2". The 'increase' being less dramatic due to incomplete knockdown is an interpretation, not a statement of fact; and the virus being 'good' at recruiting Grb2 sounds overly colloquial.*Reviewer #4:*

There are several strengths to this manuscript. The experiments are well thought out and performed, the data and analysis are high quality, and the writing is clear and concise. The data obtains from the experiments that the authors performed support their conclusions and create new and interesting questions that focus on the organization of protein oligomers that make up numerous signal transduction networks at the plasma membrane. Perhaps the most important observation is that the arrangement of Grb2 and Nck binding sites on disordered tails controls actin polymerization on more than just A36. By testing this arrangement on an unrelated viral protein, p14, the authors reinforce that the spatial positioning of Grb2 and Nck binding sites on disordered tails is a general principle that should be considered and tested when studying membrane-associated signaling systems. The major weakness of this paper lies over-extending the general principles to the field of biological phase separation. This isn't to say that the general principles revealed in their manuscript aren't applicable to the field, but the connection, as is, is overstated. Because the referenced LAT and nephrin phase separating systems are dependent on multivalency for either Grb2 or Nck, it is unclear how well A36, which contains only a single binding site for either Grb2 or Nck and has not been shown to undergo phase separation, will be predictive of LAT or nephrin condensates. However, this weakness can be easily addressed by either a deeper discussion of the implications for biological condensate formation, and function by phase separation or by toning down the assertion that there are implications for biological phase separation. Regardless of the implications for phase separation biology, this study offers a unique look into the spatial arrangement that regulates pathogen signaling and behavior and is therefore of wide interest across multiple fields of biology.

Comments:

1) 'Arp2/3' should be 'Arp2/3 complex' throughout the manuscript (it is currently written as both Arp2/3 or Arp2/3 complex).

2) In lines 99-102, the authors state that "both WIP and N-WASP only have two Nck binding sites, each with distinct preferences for the three adaptor SH3 domains." The way that this is written implies that there is a single binding configuration for the three SH3 domains of Nck for the PRMs in WIP and N-WASP which would generate heterodimers of Nck with either WIP or N-WASP. I'm not sure this is what the authors intended to say; from a biochemical perspective each SH3 domain of Nck has a different affinity for individual PRMs in WIP and N-WASP and a more likely outcome at high concentrations, like those bound at A36, would result in a network in which individual Nck SH3 domains simultaneously bind any of the numerous PRMs in WIP and N-WASP (>6 in N-WASP) to generate an Nck-WIP-N-WASP network.

3) The authors make a viral mutant that contains two Grb2 binding sites (G-G-N and G-N-G and state that their data demonstrates that the position Nck and Grb2 binding sites controls signaling output and that the number of Grb2 binding sites does not influence signaling output (Line 280, section title). While this is certainly the case for actin polymerization, it may not be accurate for other signaling pathways downstream of Grb2. To this reviewer, there are two avenues to address this: 1) Change the section title to reflect only actin polymerization, which their data supports, or 2) perform additional experiments to assess Grb2-specific signaling networks. Given that the authors have previously not observed Sos1 at Vaccinia A36 with wild-type tails (Scaplehorn et al., 2002), perhaps adding a second binding sites would result in Sos1 recruitment and alter signaling downstream of A36. The impact of the manuscript will not be lessened by changing the section title, but the impact of the manuscript will be expanded with this analysis, even if it is a bit tangential to the story.

4) There are two changes to the discussion that this reviewer would appreciate. The first is expanding the discussion of potential underlying mechanisms that would require specific positioning of Nck and Grb2 considering the data presented in the manuscript. Might the positioning of N-WASP nearer the membrane be important for N-WASP – PIP2 binding (Benesch et al., J Biol Chem 2002; Papayannopoulos et al., Mol Cell 2005; Rivera et al., Mol Cell 2009)? Or might the positioning of N-WASP, due to the spatial arrangement of Nck and Grb2 binding sites, result in better access to the existing actin cortex at the plasma membrane of the cell?

5) The second change is reducing the emphasis on the implications for membrane associated condensate formation and behavior. Given the inherent differences in valency between A36 and LAT and Nephrin, it is a little bit of a stretch to directly compare A36 with these signaling networks. However, if there is evidence of A36 undergoing a phase transition on membranes, this should be noted and emphasized. Otherwise, there is the possibility that A36 isn’t undergoing a phase transition. Instead, it could be a part of an oligomeric signaling network at the membrane. To this reviewer, this is a more interesting take on these results, as it would provide an alternative mechanism for promoting actin polymerization at the cell membrane which may be applicable to other signaling networks that control actin polymerization, such as those associated with *Listeria* or Shigella, where there is a fixed number (and density?) of receptors on the membrane that contain a single binding site for effector proteins. This isn’t to say that the authors should remove the current discussion, as it is an important consideration for the community of biological phase separation. Rather, the authors should de-emphasize this portion of their abstract and discussion because there are other incredibly interesting implications for the presented observations that can be emphasized to a greater degree.

---

## [Author Response]

Reviewer #1:Based on the model in Figure 1, it is implicit in the authors’ thinking that both NCK and GRB2 can bind simultaneously to a single diphosphorylated pY112/pY132 A36 molecule, but they provide no evidence that this is actually the case, or even evidence that both Y112 and Y132 sites are phosphorylated in a single A36 molecule (digestion of phospho-A36 with a Lys-specific protease, which will put both Y112 and Y132 in the same peptide, followed by MS analysis, perhaps with NCK or GRB2 SH2 domain pre-enrichment, might provide such evidence).

We agree this would be wonderful to know. However, we have talked to our mass spec facility and based on the sequence of A36 they have told us that the resulting peptide with both Y112 and Y132 residues would be too big to analyse in their machine. Independent of this, any mass spec analysis will examine a peptide from a population of A36 molecules so it would be impossible to know if Y112 and Y132 are both phosphorylated in a single peptide from one A36 molecule unless all A36 molecules are phosphorylated. However, we don’t know if every A36 molecule is even phosphorylated (see point 10 below). If they are also in separate peptides there is also the issue of how well the peptides fly in the mass spec. This again will cause issues in determining stoichiometry of the phosphorylation sites.

If NCK and GRB2 can indeed bind simultaneously to the same A36 molecule, this raises the question of whether the two bound proteins physically interact, and whether such an interaction might have functional consequences. Extending this idea, why then would switching the order of the two pTyr sites reduce signal output? Here, one should note that the binding of the SH2 domains of NCK and GRB2 to their target pTyr sites is directionally oriented, and by reversing the order of the sites such protein-protein interactions might not be able to take place properly. In this regard, perhaps surprisingly, the authors did not test whether the spacing between the two pTyr sites is critical (i.e. does it have to be 20 amino acids? N.B. The spacing between the two pTyr residues in ITAMs is 11 aa, and these bind tandem SH2 domains), which could be the case if specific interactions between the bound NCK and GRB2 proteins are required. Here, it is notable that the spacing of the functional NCK and GRB2 binding sites in the orthoreovirus p14 protein is also 20 amino acids. Although the authors showed that loss of GRB2 binding did not reduce NCK binding, which implies that their binding is not cooperative, it would certainly be informative to test the effects of increasing or decreasing the spacing of the NCK and GRB2 SH2 binding sites on the output of the A36 network.

We agree this is important to know and have in the past considered how to tackle this point. As the reviewer rightly points out, ITAMs can be closely spaced to facilitate binding of tandem SH2 domains in the same molecule, for example those of Syk kinase, which in turn amplifies B cell signalling. Interestingly in several cases we have noticed that the spacing between tyrosines that bind separate SH2 adaptor molecules is approximately 20 amino acids, for example: Grb2-binding Tyr171 and Tyr191 in LAT (Huang et al., 2017, PMID: 29182244), Nck-binding Tyr1176, Tyr1193, Tyr1217 in nephrin (Jones et al., 2006, PMID: 16525419) and Grb2-binding Tyr1092 and Tyr1110 in EGFR (as described in Figure 1 suppl1 of our manuscript).

To test whether this spacing is important in our signalling network, we have analyzed the impact of making the linker between the SH2 binding sites 3 times its original length. We find that actin tails induced by the new recombinant virus are still shorter when in the swopped configuration (G-N rather than N-G) (Figure 2 supplement 2B).

It might also be worth carrying out structural modeling of a diphosphorylated WT A36 105-140 aa peptide bound simultaneously to an NCK SH2 domain (plus the neighboring SH3 domain?) and a GRB2 SH2 domain (plus the neighboring SH3 domains?) to see if any structural constraints emerge, and whether any differences are apparent when the phosphorylated G-N version of this peptide is used. Such analysis might provide further insights into the preferred directionality.

There is unfortunately no structure of full length Nck. However, there is a structure (PDB 2CI9) of the SH2 domain of Nck1 bound to a phosphopeptide (EEHI-pY-DEVAADP) of Tir, which is responsible for inducing EPEC actin pedestals (Frese et al., 2006 PMID: 16636066). This Tir peptide has significant homology to the region surrounding Tyrosine112 of A36 (TEHI-pY-DSVAGSY). It is therefore likely that A36 will bind the Nck SH2 domain in a very similar fashion. There is more structural information available for Grb2 and its association with phosphotyrosine ligands. Grb2 has an exposed pY binding pocket on the SH2 domain that is well removed from its two SH3 domain, thus allowing phosphotyrosine ligand binding independent of SH3 domain engagement. Binding of ifferl ligands is, however, thought to facilitate subsequent SH3 domain interactions. For example, the association of HER2 pY with GRB2 drives SOS1 association with Grb2 nSH3 and induces a conformational change in GRB2, allowing GAB1 to access the cSH3 domain in a non-competitive manner. Based on the available information and discussions with Neil McDonald (a structural biologist at the Crick who studies RTK/RET signalling) we believe that the footprint and size of Nck1 SH2 and Grb2 SH2 binding sites for pY and domain size and orientation is consistent with space for independent binding events (no steric clash). The striking 20 amino acid spacing between phosphotyrosine SH2 binding sites in LAT, EGFR and Nephrin (see point 2 above) as well as orthoreovirus p14 also supports this notion.

We have now added an additional paragraph in the discussion in which we bring out these points including the unique nature of the Grb2 Sh2 interaction with P-tyr ligands (see section starting at line 395). Clearly we are still lacking many important structures including that of full length Nck that would significantly improve our mechanistic understanding of organization signalling networks.

Points: 1. Figure 2: Did the authors show that the GRB2 Y112 site and NCK Y122 sites in the A36 G-N protein are phosphorylated to the same extent as in A36 N-G protein? In Figure 6, they used anti-pY132 antibodies to demonstrate that phosphorylation of the pY132 GRB2 site was similar or even higher in the G-N protein than the parental N-G protein, but apparently did not do this for the pY112, perhaps due to the lack of suitable ifferl-specific antibodies. Instead, it might be possible to use GST-NCK SH2 domain and GST-GRB2 SH2 domain overlay blots of A36 N-G and G-N proteins isolated from infected cells for this purpose.

We previously tried to generate a phosphoY112 antibody, when we generated the phosphoY132 antibody (Newsome 2004) but unfortunately it did not work. Given the unpredictability of the immune response required to generate phosphoY112 antibody we have instead used GFP-Nck (SH2) recruitment in Nck-/- cells as a reporter for phosphorylation of tyrosine 112. Using only the SH2 domain ensures any recruitment must be due to phosphorylation and not any other interaction. The use of Nck-/- cells also ensures there is no competition from endogenous Nck. This new data (Figure 6C) reveals that swopping the positions of the two adaptor binding sites does not impact on the level of GFP-Nck (SH2) recruitment, which indicates Tyr112 is similarly phosphorylated in the A36 N-G and G-N viruses. Importantly, this GFP reporter is not recruited to the virus when Tyrosine 112 is mutated to Phenylalanine (Figure 6 supplement 1B). We used this direct approach of looking at the phosphoprotein at its functional site rather than the overlay blot approach suggested by the reviewer because it does not involve making cell extracts in which it would be necessary to inhibit phosphatases which may change relative phosphorylation levels.

2. Figure 3F: The authors used phosphopeptide pulldowns to show that a p14 pY96 peptide pulled down NCK and that a p14 pY116 peptide pulled down GRB2, but it appears that they did not test if a p14 pY100 peptide could pull down NCK, which seems important. Even though the Y100F mutant form of A36-p15 did not exhibit a deleterious phenotype, it is unclear whether Y100 is in fact phosphorylated in infected cells, and, more importantly, whether it can recruit NCK if it is phosphorylated, e.g. in a Y96F/Y116F mutant background. As indicated above, it is possible that the shorter 16-residue spacing between pY100 and pY116 is incompatible with appropriate signal output.

We have provided the requested pulldowns, which demonstrate the ifferl Y100 peptide does not interact with Nck or Grb2 (Figure 3 supplement 1B).

3. Figure 4: Did the authors demonstrate directly that both Y96 and Y116 in the A36-p14 N-G fusion are phosphorylated and that A36-p14 N-G binds both NCK and GRB2?

The reviewer is correct that we did not provide any data in cells confirming that phosphorylated Y96 and Y116 recruit Nck and Grb2 respectively. We have now examined the recruitment of GFP-tagged Nck and Grb2 to two new recombinant viruses in which Y96 and Y116 are mutated. These new data in (Figure 3 supplement 1A) demonstrate that loss of Y96 but not Y116 leads to a loss of Nck recruitment. Mutation of either residue leads to loss of Grb2 recruitment. We assume the loss of Grb2 recruitment in the Y96F mutant reflects a requirement for the presence of N-WASP (recruited downstream of Nck) for the presence of this adaptor as also seen with Vaccinia (Scaplehorn et al., 2002 and Weisswange et al., 2009).

4. Figure 6: The fact that siRNA knockdown of GRB2 increased pY132 antibody labeling on the A36 N-G virus is interesting. However, the authors should note that SH2 domain binding is reported to increase pTyr levels of target residues in vivo (e.g. see PMID: 1537335), presumably because the SH2-bound pTyr residues are protected from dephosphorylation by PTPs. If this is also the case for GRB2, then one might have expected the siRNA-mediated reduction in GRB2 levels to cause a decrease in pY132 levels, rather than an increase as observed. This issue would be worth some discussion.

We initially had this in the discussion of an earlier draft of the manuscript before submission but it got edited out when we were reducing the word count. We have now added the removed text and discuss its relevance in light of our observations (see lines 395 to 415).

5. Figure 7: The authors should indicate in the text that spacing of the extra GRB2 sites in A36-G-N-G and G-G-N viruses was also 20 aa.

We have indicated this in the text (see line 316).

6. Did the authors make and test an A36 N-N virus? This might generate too “strong” a signal and perturb the output. A pertinent example here would be ES cells in which the key FGFR-GRB2/SOS-RAS-ERK pathway is disrupted when endogenous GRB2 is replaced by a GRB2 with a superbinder SH2 domain that causes pathway hyperactivation and thereby prevents primitive endodermal lineage formation (PMID: 23452850).

We have generated recombinant viruses and examine the impact of having both two Nck or two Grb2 binding sites on actin tail formation. These data in (Figure 2 supplement 2A) reveals that there is no improvement in the number of virus inducing actin tails or their length when there are two binding sites for either adaptor.

7. A propos whether both Y112 and Y132 are simultaneously phosphorylated in a single A36 molecule, is the stoichiometry of phosphorylation at these two sites known. Also, would it be possible for two neighboring A36 molecules in the outer membrane of a single virion to collaborate if Y112 is phosphorylated in one A36 molecule and Y132 phosphorylated in the second A36 molecule?

The reviewer raises an interesting point, which to our knowledge has not been addressed in any system (eg phosphorylated receptors) as mass spec and ifferl-blots are bulk rather than single molecule assays. Our ongoing estimates of the number of A36 molecules on the virus, using GFPnanocages as a reference is over 500. However, the number of these molecules that are actually phosphorylated remains unknown. Independent of this issue, we would rather favour both residues are phosphorylated in a single molecule although we have no formal evidence that this is actually true. One reason to think this is we and the Kalman lab have seen that active Src/Abl family kinases are constitutively associated with virus undergoing actin dependent motility (Newsome et al., 2004. PMID: 15297625 and 2006 PMID: 16441434; Reeves et al., 2005 PMID: 15980865). This contrasts the situation in most other contexts where, Src/Abl kinases are only associated transiently to phosphorylate their substrates. This constitutive association of Src /Abl with the virus might explain why we don’t see a reduction in P-tyr signal when Grb2 is not recruited (see point 7 above). Our new data in Figure 6 A, B also reveals that swopping the position of the adaptor binding sites does not impact the levels of activated Src associated with the virus.

Reviewer #2:In my view the most serious concerns with the data as presented relate to mechanisms that might explain the apparent decrease in Grb2 binding when it is positioned N-terminal to the Nck binding site. I think the simplest explanation is that for some reason Grb2 binding sites are not as well phosphorylated when positioned N-terminally. This could be due to decreased accessibility to the kinases, increased accessibility to phosphatases, decreased accessibility of the phosphorylated site to Grb2, or many other possibilities. The experiments that show robust phosphorylation of the N-terminal Grb2 site (Figure 6) are compromised by several factors. First, only a phosphospecific antibody to the Grb2 site is used in these experiments. Similar experiments using a phosphospecific Nck binding site antibody in parallel would control for a number of issues, including whether position consistently affects relative steady state phosphorylation of both sites.

Reviewer 1 raised a similar issue (see Q4 and response above) which we have addressed using the recruitment of GFP-Nck (SH2) domain in Nck-/- cells as a reporter for phosphorylation of tyrosine 112. This new data in figure 6C confirms tyrosine 112 is equally well phosphorylated in the swopped configuration.

Furthermore, while I understand that the authors are most interested in the phosphorylation state of A36 directly under virus particles (where actin tails are assembling), it would be very informative to see results from simple immunoblots of whole cell lysates to assess bulk phosphorylation of Nck and Grb2 sites in cells infected with various viruses.

Only a small proportion of A36 is phosphorylated at the plasma membrane in infected cells as most of the protein remains in the Golgi. It was for this reason we decided to look directly at the tyrosine phosphorylated population by IF rather than indirect immunoblots. In the past we have struggled to do tyrosine phosphoblots on A36. However, after testing several different phosphotyrosine antibody and gel running conditions we now show that the level of A36 phosphorylation of A36 in N-G and G-N virus infected cells is similar (Figure 6 supplement 1A).

The argument that the apparent increase in Grb2 binding site phosphorylation might be due to decreased shielding from antibody of sites by bound Grb2 (p. 10) is not very compelling to me, especially since Grb2 knockdown has a very modest (and statistically insignificant) effect on antibody binding to this site.

We agree that the result is not as impressive as one would like. However, in many respects it is amazing that there is a difference as we know the virus is very efficient at recruiting host factors. Basically, it is very good at recruiting any residual protein after an RNAi knockdown. The other issue is we do not have an antibody to look at endogenous Grb2 localization. To get around the lack of antibody we have made use of a GFP-GRB2 stable cell line to look at recruitment (see Figure 4A and 5A).

We have now repeated our knockdown experiment with new siRNA oligos to deplete endogenous Grb2. The level of Grb2 knockdown was better although still not complete. We have replaced the original graph with the new data (Figure 6 supplement 1C) but once again the difference is not statistically significant although the ratio of the means has changed to 1.64 from 1.24 fold. As an alternative to RNAi, we also tried to use CRISPR based approaches to make a Grb2 KO HeLa cell line. Unfortunately, our HeLa cells did not survive single cell cloning or FACS sorting after transfection of three different targeting vectors.

Furthermore, the authors do not account for the effect of bound SH2 domains in increasing overall phosphorylation of their binding sites by protecting them from dephosphorylation; this would tend to oppose any apparent decrease in antibody accessibility due to SH2 binding.

This point was also raised by reviewer 1 (Q 7 and 10). We assume that our phosphoY132 antibody would only work if there was a free phosphate group that is not bound by the SH2 domain of Grb2. If this is the case, then the lack of Grb2 recruitment to the A36 G-N virus would explain why the signal of the pY132 antibody is greater than the A36 N-G virus (Figure 6D).

We have previously looked to see if the virus recruited tyrosine phosphatases (Tensin1, 2 and 3 as well as SHP1 and SHP2). We saw no evidence for recruitment. This doesn’t rule out that another tyrosine phosphatase is present in the system. However, as mentioned above (reviewer 1 Q10) the virus constitutively recruits and activates Src/Abl family kinases. In light of this, it is possible that there is little or no role for a tyrosine phosphatase in the system (an interesting notion in itself). This would also help explain why in the A36 G-N virus we see more pY132 signal (Figure 6D) and also why knockdown of Grb2 leads to a modest increase in signal (Figure 6 supplement 1C). We mention these issues/possibilities in the discussion.

In general, I think it would have been helpful to compare results for various constructs with those for the N-X virus (completely lacking the Grb2 site), in order to assess their activity relative to when Nck binding is maximal and Grb2 binding is abolished. However I don’t believe this is essential, given the amount of work required to re-do experiments.

This data was provided for Figure 5C and it is not immediately clear to us which additional experiments the reviewer would like to see or if it is more about putting the existing N-X data into graphs in the main figures? Given, this we have not added any additional data to address the reviewers question given they state it is not essential and the amount of additional work that would be required will not impact on the take home message of our study.

Reviewer #3:A key question is whether the authors are assessing exclusively the impact of the order of sites, or whether some sites are more or less efficient within the context of the full-length protein. Figure 2B Suppl 1 may indicate the authors’ interpretation is not as straightforward as it seems, all three replicates reveal a downward trend in actin tail length when comparing A36-NX with A36-X-N. While not significant according to the authors’ statistical test, this may be a question of sensitivity. This suggests that the difference between the two sites is not as black and white as the authors interpret. Conversely, when it suits the authors’ hypotheses, they choose to interpret non-significant differences as supporting their claims (increased pY132 labelling in 6B). I recommend couching the interpretation of Figure 2B Suppl 1 to include the possibility that the sites may not be equivalent or provide data that better supports the authors’ interpretation.

We have changed the text accordingly to ensure we are consistent in our language and interpretations throughout the manuscript.

Some details on the validation of recombinant viruses appears to be missing. Were multiple independent clones testing to exclude the possibility that selected viruses didn’t carry extraneous (off-target) mutations. The information that multiple clones had similar plaque sizes, for example, would strengthen the analysis.

We actually isolated two independent clones for most of the virus strains. We have now provided plaque sizes for key viruses A36 G-N (Figure 2 supplement 1C) and A36 G-N-G and A36 G-G-N (Figure 7 supplement 2A, B). These new data reveal there are no differences between the different independent virus isolates.

In general, the manuscript was clear, concise and well written. I do think the sentence on L274 should be reworded: “This increase was less dramatic than the G-N virus as the knockdown was not complete and the virus is good at recruiting any remaining Grb2”. The ‘increase’ being less dramatic due to incomplete knockdown is an interpretation, not a statement of fact; and the virus being ‘good’ at recruiting Grb2 sounds overly colloquial.

This point was raised by reviewer 2 and we have edited the text as requested in the absence of being able to generate a Grb2 KO HeLa cell line using CRISPR based approaches.

Reviewer #4 (Recommendations for the authors):There are several strengths to this manuscript. The experiments are well thought out and performed, the data and analysis are high quality, and the writing is clear and concise. The data obtains from the experiments that the authors performed support their conclusions and create new and interesting questions that focus on the organization of protein oligomers that make up numerous signal transduction networks at the plasma membrane. Perhaps the most important observation is that the arrangement of Grb2 and Nck binding sites on disordered tails controls actin polymerization on more than just A36. By testing this arrangement on an unrelated viral protein, p14, the authors reinforce that the spatial positioning of Grb2 and Nck binding sites on disordered tails is a general principle that should be considered and tested when studying membrane-associated signaling systems. The major weakness of this paper lies over-extending the general principles to the field of biological phase separation. This isn’t to say that the general principles revealed in their manuscript aren’t applicable to the field, but the connection, as is, is overstated. Because the referenced LAT and nephrin phase separating systems are dependent on multivalency for either Grb2 or Nck, it is unclear how well A36, which contains only a single binding site for either Grb2 or Nck and has not been shown to undergo phase separation, will be predictive of LAT or nephrin condensates. However, this weakness can be easily addressed by either a deeper discussion of the implications for biological condensate formation, and function by phase separation or by toning down the assertion that there are implications for biological phase separation. Regardless of the implications for phase separation biology, this study offers a unique look into the spatial arrangement that regulates pathogen signaling and behavior and is therefore of wide interest across multiple fields of biology.

We have modified the text as requested to avoid over statements.

Comments:1) ‘Arp2/3’ should be ‘Arp2/3 complex’ throughout the manuscript (it is currently written as both Arp2/3 or Arp2/3 complex).

We have modified the text so we always use Arp2/3 complex throughout the manuscript.

2) In lines 99-102, the authors state that “both WIP and N-WASP only have two Nck binding sites, each with distinct preferences for the three adaptor SH3 domains.” The way that this is written implies that there is a single binding configuration for the three SH3 domains of Nck for the PRMs in WIP and N-WASP which would generate heterodimers of Nck with either WIP or N-WASP. I’m not sure this is what the authors intended to say; from a biochemical perspective each SH3 domain of Nck has a different affinity for individual PRMs in WIP and N-WASP and a more likely outcome at high concentrations, like those bound at A36, would result in a network in which individual Nck SH3 domains simultaneously bind any of the numerous PRMs in WIP and N-WASP (>6 in N-WASP) to generate an Nck-WIP-N-WASP network.

Our original experiments with peptide arrays and recombinant Nck demonstrated that there are only two binding sites for the adapter in both WIP and N-WASP (Donnelly et al., 2013 PMID: 23707428). These experiments were performed at concentrations which are likely to be much higher than on the virus. Furthermore, our pulldowns with GFP-WIP lacking these two binding sites failed to interact with Nck although the protein was still capable of binding N-WASP via its WH1 domain (See Donnelly Figure 2B). In addition, GFP-N-WASP but not the mutant lacking the two Nck binding sites was unable to interact with Nck in the absence of WIP and its homologue WIRE (See Donnelly Figure 3C). In our mind these well controlled data demonstrate independent of the different affinities each Nck SH3 might have for PRMS that there are only 2 Nck binding sites in WIP and N-WASP. Furthermore, additional in vitro peptide pulldown experiments with recombinant Nck SH3 mutants demonstrated that the two binding sites in WIP are only recognized by the first and third Nck SH3 domains, while the first and second SH3 domains have a higher affinity for the two sites in N-WASP (See Donnelly Figure 4B). Nck with a mutated second SH3 binding site is also incapable of pulling down N-WASP in cell lysates (See Donnelly Figure 4E). Given this, we think that different SH3 sites are capable of interacting but at the end of the day there are only two Nck binding sites in WIP and N-WASP rather than the higher numbers as many people think. Moreover, I think there is a lot more to SH3 binding site specificity beyond PxxP that the field still needs to explore. We have modified the discussion to bring these points out (see text starting at line 369).

3) The authors make a viral mutant that contains two Grb2 binding sites (G-G-N and G-N-G and state that their data demonstrates that the position Nck and Grb2 binding sites controls signaling output and that the number of Grb2 binding sites does not influence signaling output (Line 280, section title). While this is certainly the case for actin polymerization, it may not be accurate for other signaling pathways downstream of Grb2. To this reviewer, there are two avenues to address this: 1) Change the section title to reflect only actin polymerization, which their data supports, or 2) perform additional experiments to assess Grb2-specific signaling networks. Given that the authors have previously not observed Sos1 at Vaccinia A36 with wild-type tails (Scaplehorn et al., 2002), perhaps adding a second binding sites would result in Sos1 recruitment and alter signaling downstream of A36. The impact of the manuscript will not be lessened by changing the section title, but the impact of the manuscript will be expanded with this analysis, even if it is a bit tangential to the story.

We have changed the title of the section so it only reflects actin polymerization.

4) There are two changes to the discussion that this reviewer would appreciate. The first is expanding the discussion of potential underlying mechanisms that would require specific positioning of Nck and Grb2 considering the data presented in the manuscript. Might the positioning of N-WASP nearer the membrane be important for N-WASP – PIP2 binding (Benesch et al., J Biol Chem 2002; Papayannopoulos et al., Mol Cell 2005; Rivera et al., Mol Cell 2009)? Or might the positioning of N-WASP, due to the spatial arrangement of Nck and Grb2 binding sites, result in better access to the existing actin cortex at the plasma membrane of the cell?

In the absence of solid structural data, including knowing how A36 is orientated relative to the membrane (Figure 1 shows it perpendicular for ease of illustration but it might be more parallel to the membrane) we feel it is difficult to speculate whether the distance between N-WASP and the plasma membrane is important. Notwithstanding this, the A36 N-X and X-N viruses (Figure 2 supplement 1B) have similar actin tail lengths even though the Nck binding site is 20 residues further away from the A36 transmembrane domain.

5) The second change is reducing the emphasis on the implications for membrane associated condensate formation and behavior. Given the inherent differences in valency between A36 and LAT and Nephrin, it is a little bit of a stretch to directly compare A36 with these signaling networks. However, if there is evidence of A36 undergoing a phase transition on membranes, this should be noted and emphasized. Otherwise, there is the possibility that A36 isn't undergoing a phase transition. Instead, it could be a part of an oligomeric signaling network at the membrane. To this reviewer, this is a more interesting take on these results, as it would provide an alternative mechanism for promoting actin polymerization at the cell membrane which may be applicable to other signaling networks that control actin polymerization, such as those associated with Listeria or Shigella, where there is a fixed number (and density?) of receptors on the membrane that contain a single binding site for effector proteins. This isn't to say that the authors should remove the current discussion, as it is an important consideration for the community of biological phase separation. Rather, the authors should de-emphasize this portion of their abstract and discussion because there are other incredibly interesting implications for the presented observations that can be emphasized to a greater degree.

As requested we have modified the abstract and discussion.